# Channel Vision Transformers: An Image Is Worth $1 \times 16 \times 16$ Words [*][†]

**Yujia Bao**[*][†]
Accenture
yujia.bao@accenture.com

**Srinivasan Sivanandan**[*]
Insitro
srinivasan@insitro.com

**Theofanis Karaletsos**[†]
Chan Zuckerberg Initiative
theofanis@karaletsos.com

## Abstract

Vision Transformer (ViT) has emerged as a powerful architecture in the realm of modern computer vision. However, its application in certain imaging fields, such as microscopy and satellite imaging, presents unique challenges. In these domains, images often contain multiple channels, each carrying semantically distinct and independent information. Furthermore, the model must demonstrate robustness to sparsity in input channels, as they may not be densely available during training or testing. In this paper, we propose a modification to the ViT architecture that enhances reasoning across the input channels and introduce Hierarchical Channel Sampling (HCS) as an additional regularization technique to ensure robustness when only partial channels are presented during test time. Our proposed model, ChannelViT, constructs patch tokens independently from each input channel and utilizes a learnable channel embedding that is added to the patch tokens, similar to positional embeddings. We evaluate the performance of ChannelViT on ImageNet, JUMP-CP (microscopy cell imaging), and So2Sat (satellite imaging). Our results show that ChannelViT outperforms ViT on classification tasks and generalizes well, even when a subset of input channels is used during testing. Across our experiments, HCS proves to be a powerful regularizer, independent of the architecture employed, suggesting itself as a straightforward technique for robust ViT training. Lastly, we find that ChannelViT generalizes effectively even when there is limited access to all channels during training, highlighting its potential for multi-channel imaging under real-world conditions with sparse sensors. Our code is available at https://github.com/insitro/ChannelViT.

## 1 Introduction

Vision Transformers (ViT) have emerged as a crucial architecture in contemporary computer vision, significantly enhancing image analysis. However, application to specific imaging domains, such as microscopy and satellite imaging, poses unique challenges. Images in these fields often comprise multiple channels, each carrying semantically distinct and independent information. The complexity is further compounded by the fact that these input channels may not always be densely available during training or testing, necessitating a model capable of handling such sparsity.

In response to these challenges, we propose a modification to the ViT architecture that bolsters reasoning across the input channels. Our proposed model, ChannelViT, constructs patch tokens independently from each input channel and incorporates a learnable channel embedding that is added to the patch tokens in addition to the location-specific positional embedding. This simple modification enables the model to reason across both locations and channels. Furthermore, by treating the

---

[*]Equal contribution.

[†]Research supporting this publication conducted while authors were employed at Insitro.

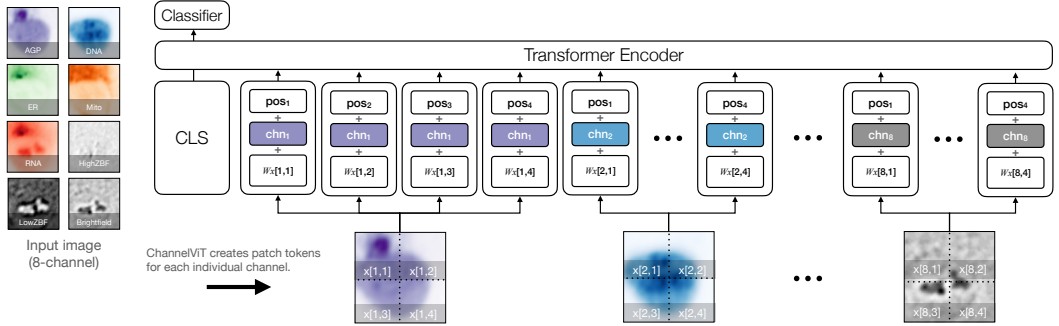

Figure 1: Illustration of Channel Vision Transformer (ChannelViT). The input for ChannelViT is a cell image from JUMP-CP, which comprises five fluorescence channels (colored differently) and three brightfield channels (colored in B&W). ChannelViT generates patch tokens for each individual channel, utilizing a learnable channel embedding chn to preserve channel-specific information. The positional embeddings pos and the linear projection $W$ are shared across all channels.

channel dimension as the patch sequence dimension, ChannelViT can seamlessly handle inputs with varying sets of channels.

Despite these advancements, two main challenges persist. While ChannelViT can leverage existing efficient implementations of ViT with minimal modifications, the increase in sequence length introduces additional computational requirements. Moreover, if ChannelViT is consistently trained on the same set of channels, its ability to generalize to unseen channel combinations at test time may be compromised. To address these challenges, we introduce Hierarchical Channel Sampling (HCS), a new regularization technique designed to improve robustness. Unlike channel dropout, which drops out each input channel independently, HCS uses a two-step sampling procedure. It first samples the number of channels and then, based on this, it samples the specific channel configurations. While channel dropout tends to allocate more distribution to combinations with a specific number of channels, HCS assigns a uniform weight to the selection of any number of channels. HCS consistently improves robustness when different channels are utilized during testing in both ViT and ChannelViT. Notably, our evaluation on ImageNet shows that using only the red channel, HCS can increase the validation accuracy from 29.39 to 68.86.

We further evaluate ChannelViT on two real world multi-channel imaging applications: microscopy cell imaging (JUMP-CP) and satellite imaging (So2Sat). In these applications, different channels often correspond to independent information sources. ChannelViT significantly outperforms its ViT counterpart in these datasets, underscoring the importance of reasoning across different channels. Moreover, by treating different channels as distinct input tokens, we demonstrate that ChannelViT can effectively generalize even when there is limited access to all channels in the dataset during training. Lastly, we show that ChannelViT enables additional insights. The learned channel embeddings correspond to meaningful interpretations, and the attention visualization highlights relevant features across spatial and spectral resolution, enhancing interpretability. This highlights the potential of ChannelViT for wide-ranging applications in the field of multi-channel imaging.

## 2 RELATED WORK

**Vision transformer and its applications to multi-channel imaging**  Vision Transformer (ViT) has demonstrated state-of-the-art performance in various computer vision tasks Dosovitskiy et al.; Touvron et al. (2021); Carion et al. (2020); Zhu et al. (2020b). Recently, researchers have started adopting ViT for multi-spectral imaging. For example, in satellite imaging, Kaselimi et al. (2022) showed that a ViT-based classifier outperforms CNN models, especially on imbalanced classes. Additionally, Tarasiou et al. (2023) proposed acquisition-time-specific temporal positional encodings to model satellite images over time, while Cong et al. (2022) demonstrated the benefits of using distinct spectral positional encodings with ViT. Recently, Nguyen et al. (2023) proposed a modification to the ViT architecture, introducing variable tokenization and variable token aggregation methods to handle heterogeneous input data sources in climate and weather modeling. Moreover, Scheibenreif

et al. (2022) found that ViT, when combined with self-supervised pre-training, performs on-par with state-of-the-art benchmarks.

In the field of cell biology, Sivanandan et al. (2023) utilized ViT with self-supervised pre-training to learn representations of cells across multiple fluorescence channels. Furthermore, Hatamizadeh et al. (2022a;b) leveraged ViT for segmenting 3D MRI images. Hussein et al. (2022) proposed to train multiple ViTs, one for each input channel, for epileptic seizure predictions.

In contrast to previous work, we address a practical challenge in multi-channel imaging, where different datasets often have different available channels.[*] To tackle this challenge, we propose ChannelViT, which creates image patches from each individual input channel. This simple modification unifies the modeling across data with different input channels and offers robust performance at test time, even when only a subset of the channels is available.

**Robustness for Vision Transformer** Robustness can be defined in different ways. One aspect is the vulnerability to adversarial attacks. Mahmood et al. (2021) found that ViTs are as susceptible to white-box adversarial attacks as CNNs. To improve robustness, Robust ViT incorporates more robust components like global pooling (Mao et al., 2022). Additionally, Chefer et al. (2022) propose regularization of the relevancy map of ViT to enhance robustness. Zhou et al. (2022); Zhang et al. (2021); Song et al. (2022) augments transformers with feature-wise attention to improve robustness and performance. Another approach focuses on generalization over distribution shifts Sagawa et al. (2019); Liu et al. (2021). Bao & Karaletsos (2023) introduces a context token inferred from ViT's hidden layers to encode group-specific information.

In our work, we specifically focus on improving the generalization performance across different channel combinations, which is a common scenario in multi-channel imaging. We argue that the original ViT is sensitive to changes in input channels, as it computes a single patch token across all channels. In contrast, ChannelViT creates separate patch tokens for each channel, making it inherently more robust to variations in channel availabilities. To further enhance channel robustness, we introduce hierarchical channel sampling (HCS) during training. This methodology draws inspiration from prior studies on channel dropout Srivastava et al. (2014); Tompson et al. (2015); Hou & Wang (2019). However, instead of dropping out intermediate channels, our approach introduces a two-stage sampling algorithm designed to selectively mask out the input channels.

## 3 METHOD

ChannelViT is a modification of the original Vision Transformer (ViT) architecture proposed by Dosovitskiy et al.. Unlike the original architecture, which condenses each multi-channel image patch into a single 'word' token, ChannelViT segregates channel-specific information into multiple tokens. This simple yet effective modification yields three key advantages:

1. ChannelViT facilitates reasoning across both positions and channels with Transformer;
2. By transforming the channel dimension into the sequence length dimension, ChannelViT can seamlessly manage inputs with varying sets of channels;
3. ChannelViT can utilize existing efficient implementations of ViT.

In the following paragraphs, we explore the architecture and implementation of ChannelViT in detail. Figure 1 provides a visual overview of the model.

### 3.1 CHANNEL VISION TRANSFORMER (CHANNELVIT)

**Patch embeddings** Consider an input image $x$ with dimensions $H \times W \times C$. Given a patch size of $P \times P$, this image can be reshaped into a sequence of non-overlapping patches

$$[x[c_1, p_1], \ldots, x[c_1, p_N], x[c_2, p_1], \ldots, x[c_2, p_N], \quad \ldots \quad, x[c_C, p_N], \ldots, x[c_C, p_N]],$$

where $x[c_i, p_n]$ corresponds to the $n$-th $P \times P$ image patch at channel $c_i$ and $N = HW/P^2$. As the Transformer encoder requires a sequence of one-dimensional vectors, each patch is flattened into a

---

[*]For example (https://github.com/chrieke/awesome-satellite-imagery-datasets), satellite imaging often involves multiple signals such as Sentinel-1 (SAR), Sentinel-2, UAV, etc.

1D vector. Unlike ViT, which generates a single token for a multi-channel image patch, ChannelViT produces one token from every single-channel image patch.

**Tied image filters**    We apply a learnable linear projection $W \in \mathbb{R}^{P^2 \times D}$ to the flattened patches. It is important to note that in a regular ViT, each channel has its own weights in the linear projection layer. In ChannelViT, our preliminary experiments suggest that tying the image filters across channels offer superior performance compared to untied image filters (Appendix C.3). Therefore, we tie the learnable projection $W$ across channels. The intuition behind this is that the low-level image filters can be shared across channels (Ghiasi et al., 2022), and tying the parameters can improve the model's robustness across channels.

**Channel-aware and position-aware patch embeddings**    Despite tying the linear filter across channels, it remains essential to preserve channel-specific information, given the distinct characteristics of different channels (Appendix C.4). We introduce learnable channel embeddings $[\text{chn}_1, \ldots, \text{chn}_C]$, where $\text{chn}_c \in \mathbb{R}^D$. In line with the original ViT, we also incorporate learnable positional embeddings to maintain positional information of each patch. We denote the positional embeddings as $[\text{pos}_1, \ldots, \text{pos}_N]$, where $\text{pos}_n \in \mathbb{R}^D$. It's worth noting that these position embeddings are also shared across channels, enabling ChannelViT to recognize the same image patch across different channels. Finally, we prepend a learnable classifier token $\text{CLS} \in \mathbb{R}^D$ to the sequence to encode global image features. The resulting input sequence can be written as

$$\big[\text{CLS}, \quad \text{pos}_1 + \text{chn}_1 + W x[c_1, p_1], \quad \ldots, \quad \text{pos}_N + \text{chn}_1 + W x[c_1, p_N],$$
$$\ldots, \quad \text{pos}_1 + \text{chn}_C + W x[c_C, p_1], \quad \ldots, \quad \text{pos}_N + \text{chn}_C + W x[c_C, p_N]\big].$$

**Transformer encoder**    Following the original VIT, we feed the above input sequence into a Transformer encoder, which captures dependencies between image patches by embedding each patch based on its similarity to others Vaswani et al. (2017). Specifically, the Transformer encoder comprises alternating layers of multiheaded self-attention blocks and MLP blocks. Layer normalization, as proposed by Ba et al. (2016), is performed before each block, and residual connections He et al. (2016) are established after each block. We use the final layer representation of the $\text{CLS}$ token to represent the input image. For classification tasks, a linear classifier is employed, followed by a Softmax function, to predict the corresponding label. We utilize the standard cross entropy loss as our training objective.

### 3.2 Hierarchical channel sampling (HCS)

Training ChannelViT directly presents two challenges: 1) The sequence length becomes proportional to the number of channels, leading to a quadratic surge in the number of attentions required for computation; 2) Training exclusively on all channels may result in the model not being prepared for partial channels at test time, thereby affecting its generalization capability. To mitigate these issues, we propose applying hierarchical channel sampling (HCS) during the training process. Specifically, for an image $x$ with $C$ channels, we proceed as follows:

1. First, we sample a random variable $m$ uniformly from the set $\{1, 2, \ldots, C\}$. This $m$ represents the number of channels that we will utilize during this training step;

2. Next, we sample a channel combination $\mathcal{C}_m$ uniformly from all channel combinations that consist of $m$ channels;

3. Finally, we return the image with only the sampled channels $x[\mathcal{C}_m]$.

HCS shares similarity to channel dropout Tompson et al. (2015), but it differs in terms of the prior distribution imposed on the sampled channels. In channel dropout, each channel is dropped based on a given probability independently. The probability of having $m$ channels varies drastically for different $m$s, which can negatively impact the final performance (Figure 3). In contrast, since $m$ is sampled uniformly over the total number of channels, HCS ensures that the sampling procedure equally covers each $m$. Finally, we note that HCS is only employed during training. At test time, ChannelViT has access to all input channels.

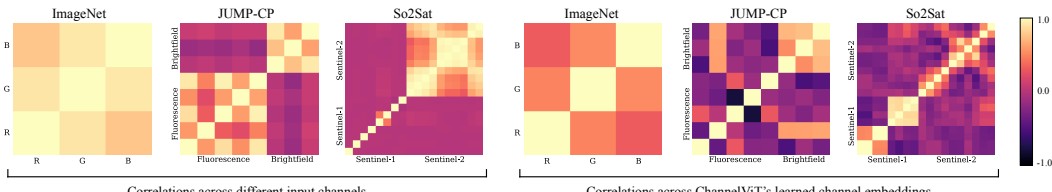

Figure 2: Correlation patterns among image channels (left) and the learned channel embeddings (right) for ImageNet, JUMPCP, and So2Sat. ImageNet displays a strong correlation among the three RGB input channels while JUMPCP and So2Sat show minimal correlation between different signal sources (Fluorescence vs. Brightfield, Sentinel 1 vs Sentinel 2).

HCS can also be interpreted as simulating test-time distributions during training. Compared to group distributionally robust optimization (Sagawa et al., 2019), HCS minimizes the mean loss rather than the worst-case loss. This approach is logical when considering channel robustness, as having more channels will naturally enhance performance. We don't want the model to over-focus on the worst-case loss, which typically corresponds to situations when we sample very few channels.

# 4 EXPERIMENTS

We evaluate ChannelViT across three image classification benchmarks: ImageNet Deng et al. (2009), JUMP-CP Chandrasekaran et al. (2022), and So2Sat Zhu et al. (2019). In Figure 2 (top), we illustrate the correlation among different input channels for each dataset. As observed, ImageNet exhibits a strong correlation among the three RGB channels. For JUMP-CP, while there is a strong correlation within the fluorescence channels and within the brightfield channels, there is minimal to no correlation between the brightfield and the fluorescence channels. A similar group structure among the channels is observed for So2Sat. Due to space constraints, our primary focus in the main paper is on the comparison between ViT and ChannelViT. For additional comparisons with MultiViT (Hussein et al., 2022), please refer to Appendix B.1. Comparisons with FANs (Zhou et al., 2022) can be found in Appendix B.2.

**JUMP-CP** The JUMP-CP benchmark, established by the JUMP-Cell Painting Consortium, serves as a microscopy imaging standard. In alignment with the work of Chandrasekaran et al. (2022), we utilize the task of perturbation detection from cell images as a means to evaluate and compare the efficacy of various representation models. It is important to recognize that while perturbation detection is a valuable task, it is not the ultimate objective of cell imaging modeling; rather, it provides an interpretable metric for model assessment. The dataset includes a total of 160 perturbations. We focused on a compound perturbation plate 'BR00116991', which contains 127k training images, 45k validation images, and 45k testing images. Each cell image contains 8 channels, comprising both fluorescence information (first five channels) and brightfield information (last three channels).

**So2Sat** This satellite imaging benchmark encompasses half a million image patches from Sentinel-1 and Sentinel-2 satellites, distributed across 42 global urban agglomerations. Each image patch incorporates 18 channels, with 8 originating from Sentinel-1 and the remaining 10 from Sentinel-2. The primary objective of this dataset is to facilitate the prediction of the climate zone for each respective image patch, with a total of 17 distinct climate zones being represented.

**Implementation details** We utilize the Vision Transformer (ViT) implementation provided by Facebook Research[†]. During training, we minimize the cross entropy loss. To ensure a fair comparison, both ViT and ChannelViT are subjected to identical optimization settings. These settings encompass the use of the Adam optimizer, a learning rate scheduler featuring linear warmup and cosine decay, and a cosine scheduler for the weight decay parameter. **For a more detailed description of the hyper-parameter settings, we direct readers to the Appendix.**

Table 1: Validation accuracy on ImageNet under different testing conditions (using all three channels or only one channel). We observe that 1) hierarchical channel sampling significantly boosts single-channel performance at test time; 2) ChannelViT consistently outperforms the ViT baseline. The expert models, trained using only one channel, represent the upper bound of potential performance.

| Backbone | Use hierarchical channel sampling? | Val Acc. on RGB | Val Acc. on R-only | Val Acc. on G-only | Val Acc. on B-only |
|---|---|---|---|---|---|
| *Models trained on three channels (RGB)* | | | | | |
| ViT-S/16 | ✗ | 71.49 | 29.39 | 33.79 | 21.18 |
| ViT-S/16 | ✓ | 73.01 | 68.86 | 69.78 | 67.59 |
| ChannelViT-S/16 | ✓ | **74.64** | **69.90** | **70.30** | **68.48** |
| *Expert models trained on only one channel* | | | | | |
| ViT-S/16 (R-only) | N/A | — | 70.04 | — | — |
| ViT-S/16 (G-only) | N/A | — | — | 70.61 | — |
| ViT-S/16 (B-only) | N/A | — | — | — | 69.47 |

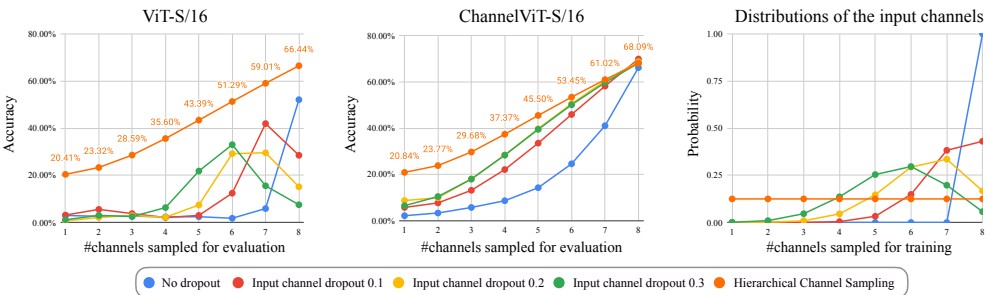

Figure 3: HCS vs. input channel dropout on JUMP-CP (trained on all 8 channels). On the left, we present the accuracy of ViT-S/16 and ChannelViT-S/16 under varying input channel dropout rates and HCS. The accuracy is evaluated across all channel combinations, with the mean accuracy reported for combinations with an equal number of channels (represented on the horizontal axis). On the right, we illustrate the probability distribution of the sampled channel combinations during the training process. We observe 1) ViTs trained with input channel dropout tend to favor channel combinations that are sampled the most; 2) ChannelViT with input channel dropout outperforms ViT with input channel dropout; 3) HCS surpasses input channel dropout in terms of channel robustness.

## 4.1 IMAGENET

Table 1 showcases our results on ImageNet, using ViT small as the representation backbone and a patch size of 16 by 16. We observe that without applying hierarchical channel sampling, ViT-S/16 achieves a validation accuracy of 71.49 using all three channels but fails to generalize when only one channel is provided at test time. Simulating this test-time channel drop during training via hierarchical channel sampling (HCS) significantly improves performance. For instance, the validation accuracy for using only the red channel improves from 29.39 to 68.86, demonstrating the effectiveness of HCS as a regularizer for enforcing channel robustness. Lastly, while there is limited room for improvement due to the strong correlations among the input RGB channels, ChannelViT still consistently outperforms the corresponding ViT baseline (by 1.2 on average), narrowing the gap $(1.30 \rightarrow 0.48)$ to the expert models that are trained using only one channel.

## 4.2 JUMP-CP: MICROSCOPY CELL IMAGING

We present our results on the microscopy cell imaging benchmark, JUMP-CP, in Table 2. This benchmark involves a 160-way classification task. Due to computational constraints, we utilize ViT-S as our representation backbone. We consider both the standard resolution with a patch size of 16x16 and a high-resolution model with a patch size of 8x8.

---

[†]https://github.com/facebookresearch/dino/blob/main/vision_transformer.py

Table 2: Test accuracy of 160-way perturbed gene prediction on JUMP-CP. Two training settings are considered: one using only 5 fluorescence channels and the other incorporating all 8 channels, which includes 3 additional brightfield channels. During testing, all possible channel combinations are evaluated and we report the mean accuracies for combinations with the same number of channels (See Appendix B for detailed error analyses). We observe that cross channel reasoning is crucial when the inputs have independent information (fluorescence vs. brightfield).

| | ViT-S/16 | ChannelViT-S/16 | ViT-S/16 | ChannelViT-S/16 | ViT-S/8 | ChannelViT-S/8 |
|---|---|---|---|---|---|---|
| Use hierarchical channel sampling? | ✗ | ✗ | ✓ | ✓ | ✓ | ✓ |
| *Training on 5 fluorescence channels* | | | | | | |
| 5 channels | 48.41 | 53.41 | 55.51 | 56.78 | **60.29** | 60.03 |
| 4 channels | 0.85 | 15.13 | 43.59 | 45.94 | 48.80 | **49.34** |
| 3 channels | 1.89 | 5.12 | 33.14 | 35.45 | 37.13 | **38.15** |
| 2 channels | 1.46 | 1.22 | 25.24 | 26.57 | 27.40 | **27.99** |
| 1 channel | 0.54 | 1.25 | 20.49 | 21.43 | 21.30 | **21.58** |
| *Training on all 8 channels (5 fluorescence channels & 3 brightfield channels)* | | | | | | |
| 8 channels | 52.06 | 66.22 | 56.87 | 68.09 | 66.44 | **74.77** |
| 7 channels | 5.91 | 41.03 | 49.35 | 61.02 | 59.01 | **68.42** |
| 6 channels | 1.81 | 24.57 | 42.38 | 53.45 | 51.29 | **61.26** |
| 5 channels | 2.46 | 14.20 | 35.78 | 45.50 | 43.39 | **53.05** |
| 4 channels | 2.38 | 8.56 | 29.84 | 37.37 | 35.60 | **43.87** |
| 3 channels | 2.70 | 5.65 | 24.94 | 29.68 | 28.59 | **34.19** |
| 2 channels | 2.63 | 3.24 | 21.54 | 23.77 | 23.32 | **25.73** |
| 1 channel | 3.00 | 2.08 | 19.92 | 20.84 | 20.41 | **21.20** |

In the first part of our analysis, we train all models using only the five fluorescence channels and evaluate their performance on the test set under various input channel combinations. Our observations are as follows: 1) HCS significantly enhances the channel robustness for both ViT and ChannelViT; 2) High-resolution models consistently outperform their low-resolution counterparts; 3) With the exception of the 5-channel evaluation with a patch size of 8x8, ChannelViT consistently outperforms ViT.

In the latter part of our analysis, we utilize all available channels for training, which includes three additional brightfield channels for each image. For ViT, the high-resolution ViT-S/8 model improves from 60.29 to 66.44, demonstrating the importance of the additional brightfield information, while the improvement for ViT-S/16 is marginal (from 55.51 to 56.87). When focusing on ChannelViT, we observe a significant performance boost over its ViT counterpart. ChannelViT-S/16 outperforms ViT-S/16 by 11.22 (68.09 vs 56.87) and ChannelViT-S/8 outperforms ViT-S/8 by 8.33 (74.77 vs. 66.44). These improvements are consistent across different channel combinations. As we have seen in Figure 2, fluorescence and brightfield channels provide distinct information. ChannelViT effectively reasons across channels, avoiding the need to collapse all information into a single token at the first layer, thereby enhancing performance.

Lastly, we delve into a comparative analysis between input channel dropout and hierarchical channel sampling, as depicted in Figure 3. It is evident from our observations that the ViT model, when trained with HCS, consistently surpasses the performance of those trained with input channel dropout across all channel combinations. Furthermore, we discern a pronounced correlation between the performance of models trained with input channel dropout and the probability distribution of the number of channels sampled during training.

**Data Efficiency**  In the realm of microscopy imaging, we often encounter situations where not all channels are available for every cell due to varying experiment guidelines and procedures. Despite this, the goal remains to develop a universal model capable of operating on inputs with differing channels. ChannelViT addresses this issue by treating different channels as distinct input tokens, making it particularly useful in scenarios where not all channels are available for all data. Table 3 presents a scenario where varying proportions (0%, 25%, 50%, 75%, 100%) of the training data have access to all eight channels, with the remaining data only having access to the five fluorescence

Table 3: ViT vs. ChannelViT when we have varying channel availability during training. Both models are trained using HCS. The accuracy is evaluated using five fluorescence channels (top) and all eight channels (bottom). For the middle three columns where we have mixed training data, we report the mean (and std) accuracy over three randomly generated partitions. ChannelViT consistently outperforms ViT across all settings, and the performance gap notably widens as access to more 8-channel data is provided.

| | Combine fluorescence-only data and 8-channel data for training | | | | |
|---|---|---|---|---|---|
| % fluorescence-only data | 100% | 75% | 50% | 25% | 0% |
| % 8-channel data | 0% | 25% | 50% | 75% | 100% |
| *Evaluating on 5 fluorescence channels* | | | | | |
| ViT-S/16 | 55.51 | $52.55_{\pm2.68}$ | $51.65_{\pm2.14}$ | $49.53_{\pm1.39}$ | 45.75 |
| ChannelViT-S/16 | **56.78** | $\mathbf{58.01}_{\pm1.77}$ | $\mathbf{58.19}_{\pm1.49}$ | $\mathbf{58.42}_{\pm1.37}$ | **57.60** |
| *Evaluating on all 8 channels* | | | | | |
| ViT-S/16 | — | $50.29_{\pm1.93}$ | $52.47_{\pm1.82}$ | $54.64_{\pm1.01}$ | 56.87 |
| ChannelViT-S/16 | — | $\mathbf{57.97}_{\pm1.36}$ | $\mathbf{61.88}_{\pm0.91}$ | $\mathbf{64.80}_{\pm0.89}$ | **68.09** |

channels. The performance of ViT and ChannelViT is evaluated at test time using both the five fluorescence channels (top section) and all eight channels (bottom section).

Our observations are as follows: 1) When only a limited amount of 8-channel data (25%) is available, both ChannelViT and ViT show a decrease in performance when utilizing eight channels at test time compared to five channels; 2) As the availability of 8-channel data increases, the performance of the ViT baseline on the fluorescence evaluation steadily declines (from 55.51 to 45.75), while the performance of ChannelViT sees a slight improvement (from 56.78 to 57.60); 3) When evaluated on all eight channels, ChannelViT significantly outperforms ViT, with an average gap of 9.62.

**Channel-specific attention visualization**    Attention heatmaps, generated by Vision Transformers (ViTs), have emerged as a valuable tool for interpreting model decisions. For instance, Chefer et al. (2021) introduced a relevance computation method, which assigns local relevance based on the Deep Taylor Decomposition principle and subsequently propagates these relevance scores through the layers. However, a limitation of ViTs is their tendency to amalgamate information across different channels. In the realm of microscopy imaging, discerning the contribution of each fluorescence channel to the predictions is vital due to their distinct biological implications.

Figure 4 (right) presents the class-specific relevance visualizations for ViT-S/8 and ChannelViT-S/8. For the top cell labeled KRAS, ChannelViT appears to utilize information from the Mito channel. For the bottom cell labeled KCNH76, ChannelViT seems to utilize information from the ER and RNA channels for its prediction. Compared to ViT, ChannelViT facilitates the examination of contributions made by individual channels.

In Figure 4 (left), we further compute the maximum attention score (averaged over 100 cells) for each cell label (perturbed gene) and each input channel. Our observations indicate that ChannelViT focuses on different channels for different labels (corresponding to perturbed genes), with the Mito channel emerging as the most significant information source. This heatmap, which describes the discriminability of different labels over different channels, can also aid in better understanding the relationships between different gene perturbations.

**Time efficiency**    One limitation of ChannelViT is the additional computational cost incurred when expanding the channel dimension into the sequence length dimension. Implementing ChannelViT without HCS increases the training time from approximately 3 hours to 12 hours. With HCS, the training duration for ChannelViT is reduced to about 10 hours. During inference, ChannelViT requires approximately 1.6 times more time than its ViT counterpart. An interesting future direction would be to combine ChannelViT with more efficient attention mechanisms, such as Linformer Wang et al. (2020) and LongNet Ding et al. (2023), which scale linearly with sequence length. We direct the reader to Appendix C.1 for a comprehensive analysis of the running times.

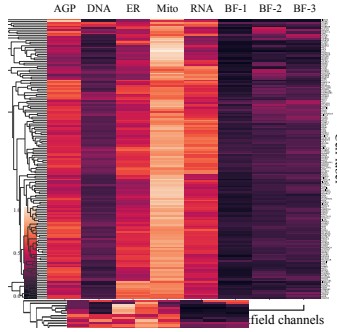
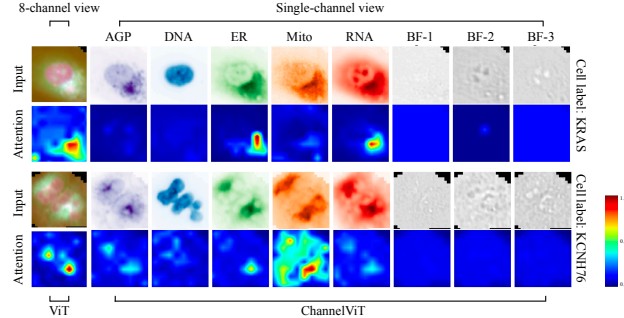
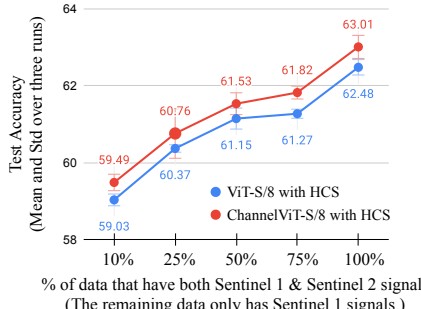

Figure 4: Left: Class-specific relevance attribution of ChannelViT-S/8 for each cell label (perturbed gene) on JUMP-CP. For each perturbed gene (y-axis) and each channel (x-axis), we calculate the maximum attention score, averaged over 100 cells from that specific cell label. This reveals that ChannelViT focuses on different input channels depending on the perturbed gene. Right: A visualization of the relevance heatmaps for both ViT-S/8 (8-channel view) and ChannelViT-S/8 (single-channel view). Both models are trained on JUMP-CP using HCS across all 8 channels. ChannelViT offers interpretability by highlighting the contributions made by each individual channel.

Table 4: Test accuracy of 17-way local climate zone classification on So2Sat. We consider two official splits: random split and city split. Both ViT and ChannelViT are trained on all channels with hierarchical channel sampling. We evaluate their performance on 18 channels (Sentinel 1 & 2) as well as partial channels (Sentinel 1).

|  | Sentinel 1 (Channel 0-7) | Sentinel 1 & 2 (Channel 0-17) |
| --- | --- | --- |
| *Random split (Zhu, 2021)* | | |
| ViT-S/8 | 50.62 | 97.82 |
| ChannelViT-S/8 | **59.75** | **99.10** |
| *City split (Zhu et al., 2019)* | | |
| ViT-S/8 | 41.07 | 62.48 |
| ChannelViT-S/8 | **47.39** | **63.01** |

Figure 5: Test accuracy on So2Sat city split with varying channel availabilities during training. Both ViT and ChannelViT are trained with hierarchical channel sampling. Performances are evaluated using all channels (Sentinel 1 & 2).

### 4.3 So2Sat: Satellite Imaging

Our results on the So2Sat satellite imaging benchmark are presented in Table 4. We evaluate two official splits: random split and city split, training both ViT-S/8 and ChannelViT-S/8 models using hierarchical channel sampling across all channels (Sentinel 1 & 2).

Upon evaluation, ChannelViT demonstrates superior performance over its ViT counterpart, with an improvement of 1.28 for the random split and 0.53 for the more challenging city split. In the realm of satellite imaging, Sentinel 1 channels are derived from a Synthetic Aperture Radar operating on the C-band, while Sentinel-2 is a multispectral high-resolution imaging mission. It's worth noting that Sentinel-2 data can be cloud-affected, underscoring the importance of models that can robustly operate under partial signals using only Sentinel 1. In both random and city splits, ChannelViT significantly outperforms ViT (59.75 vs. 50.62 in random split and 47.39 vs. 41.07 in city split).

Lastly, we explore the efficiency of ChannelViT in combining satellite training data with different signals. As depicted in Figure 5, we consider varying proportions (10%, 25%, 50%, 75%, 100%) of the training data with access to both Sentinel 1 & 2 signals, while the remaining data only has access to Sentinel 1 signals. The models are evaluated using all Sentinel 1 & 2 signals. Our observations consistently show ChannelViT outperforming ViT.

**Interpreting the channel embeddings learned by ChannelViT** Figure 2 presents the correlations between the input channels. It's noteworthy that the first four channels of Sentinel-1 corre-

spond to: 1) the real part of the VH channel; 2) the imaginary part of the VH channel; 3) the real part of the VV channel; and 4) the imaginary part of the VV channel. These four input channels are uncorrelated, as evidenced by the bottom left corner of the So2Sat visualization heatmap. However, upon examining the correlations between the learned channel embeddings, we observe a high correlation between the real and imaginary parts of both VV and VH channels. This intuitively aligns with the fact that the real and imaginary parts are equivalent in terms of the information they provide. This demonstrates that ChannelViT learns meaningful channel embeddings, which can provide additional insights into the relationships between different input signals.

## 5 CONCLUSION

In conclusion, our proposed model, ChannelViT, effectively addresses the unique challenges of multi-channel imaging domains. By enhancing reasoning across input channels and seamlessly handling inputs with varying sets of channels, ChannelViT has consistently outperformed its ViT counterpart in our evaluations on ImageNet and diverse applications such as medical, microscopy cell, and satellite imaging. The introduction of Hierarchical Channel Sampling (HCS) further bolsters the model's robustness when testing with different channel combinations. Moreover, ChannelViT not only improves data efficiency but also provides additional interpretability, underscoring its potential for broad applications in the field of multi-channel imaging.

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

APPENDIX

# A    IMPLEMENTATION DETAILS

This section elucidates the specifics of our implementation and the settings of our hyper-parameters.

## A.1    HIERARCHICAL CHANNEL SAMPLING

In Section 3.2, we outlined the channel sampling procedure of HCS. In this subsection, we offer a comprehensive example of HCS in conjunction with ChannelViT and ViT.

**Hierarchical Channel Sampling for ChannelViT**    Given a three-channel input $x$, as per Section 3.1, the input sequence for the Transformer encoder can be expressed as

$$\big[\texttt{CLS}, \quad \text{pos}_1 + \text{chn}_1 + Wx[c_1, p_1], \quad \ldots, \quad \text{pos}_N + \text{chn}_1 + Wx[c_1, p_N],$$
$$\text{pos}_1 + \text{chn}_2 + Wx[c_2, p_1], \quad \ldots, \quad \text{pos}_N + \text{chn}_2 + Wx[c_2, p_N],$$
$$\text{pos}_1 + \text{chn}_3 + Wx[c_3, p_1], \quad \ldots, \quad \text{pos}_N + \text{chn}_3 + Wx[c_3, p_N]\big].$$

Let's assume that our sampled channel combination from the HCS algorithm is $\{1, 3\}$. The corresponding input sequence for the Transformer encoder would then be modified accordingly.

$$\big[\texttt{CLS}, \quad \text{pos}_1 + \text{chn}_1 + Wx[c_1, p_1], \quad \ldots, \quad \text{pos}_N + \text{chn}_1 + Wx[c_1, p_N],$$
$$\text{pos}_1 + \text{chn}_3 + Wx[c_3, p_1], \quad \ldots, \quad \text{pos}_N + \text{chn}_3 + Wx[c_3, p_N]\big].$$

It's important to note that reducing the number of channels only modifies the sequence length. Furthermore, since we sample the channel combinations for each training step, the channels utilized for each image can vary across different epochs.

**Hierarchical Channel Sampling for ViT**    Given the identical three-channel input $x$, the input sequence for the Transformer encoder can be articulated as

$$\big[\texttt{CLS}, \quad \text{pos}_1 + W_1 x[c_1, p_1] + W_2 x[c_2, p_1] + W_3 x[c_3, p_1] + b,$$
$$\ldots, \quad \text{pos}_n + W_1 x[c_1, p_n] + W_2 x[c_2, p_n] + W_3 x[c_3, p_n] + b\big].$$

Here $W_1, W_2, W_3$ represent the weights associated with each input channel, and $b$ is the bias term. Let's continue with the assumption that our sampled channel combination from the HCS algorithm remains $\{1, 3\}$. We then adjust the above input sequence as follows:

$$\big[\texttt{CLS}, \quad \text{pos}_1 + W_1 x[c_1, p_1]3/2 + W_3 x[c_3, p_1]3/2 + b,$$
$$\ldots, \quad \text{pos}_n + W_1 x[c_1, p_n]3/2 + W_3 x[c_3, p_n]3/2 + b\big].$$

It's noteworthy that, in addition to masking the input from the second channel, we also rescale the remaining channels by a factor of $3/2$. This is akin to the approach of Srivastava et al. (2014), and is done to ensure that the output of the linear patch layer maintains the same scale, despite the reduction in input channels.

## A.2    TRAINING WITH VIT AND CHANNELVIT

**Backbone**    For the vision transformer backbone, we employ the PyTorch implementation provided by Facebook Research[‡]. Due to computational constraints, we primarily utilize the 'vit-small' architecture, which has an embedding dimension of 386, a depth of 12, 6 heads, an MLP hidden dimension of $4 \times 386 = 1544$ and pre layer normalization. We also briefly experiment 'vit-base' which increases the embedding dimension to 768, the number of heads to 12, and the MLP hidden dimension to $4 \times 768 = 3072$. For ChannelViT, we retain the same parameter settings as its ViT counterparts for the Transformer encoder. Note that ChannelViT has a marginally smaller number of parameters, as the first linear projection layer is now shared across channels.

**Objective**    We employ the standard cross-entropy loss for both ViT and ChannelViT across the four image classification benchmarks. Specifically, we utilize the Transformer encoder's representation for the CLS token at the final layer, and append a linear layer, followed by a Softmax function, to predict the probability of each class.

---

[‡] https://github.com/facebookresearch/dino/blob/main/vision_transformer.py

**Optimization**   For optimization, we employ the AdamW optimizer (Loshchilov & Hutter, 2019). The learning rate is warmed up for the initial 10 epochs, peaking at 0.0005 (Goyal et al., 2017), after which it gradually decays to $10^{-6}$ following a cosine scheduler. To mitigate overfitting, we apply weight decay to the weight parameters, excluding the bias and normalization terms. The weight decay starts at 0.04 and incrementally increases during training, following a cosine scheduler, up to a maximum of 0.4. Each model is trained for 100 epochs with a batch size of 256. The training is conducted on an AWS `p4d.24xlarge` instance equipped with 8 A100 GPUs.

### A.3   TRAINING ON DATASETS WITH VARYING CHANNEL AVAILABILITY

In Table 3 and Figure 5, we investigated scenarios where our training datasets exhibited varying channel availability. This section provides a detailed description of the training settings we employed and presents additional results for an alternative setting.

**ChannelViT and ViT**   Despite the different channel combinations in the training datasets, we utilize a consistent approach (as detailed in Appendix ) to encode the images for both ChannelViT and ViT. For ChannelViT, this entails having varying sequence lengths for images with different numbers of channels. For ViT, this involves masking out the unavailable channels and rescaling the remaining ones.

**Objective**   We continue to use the cross-entropy loss. However, in this instance, there are two potential methods for data sampling.

1. Sampling a random batch from each dataset and minimizing their average loss. This approach will assign more weight to datasets with fewer examples. Mathematically, it optimizes

$$\mathcal{L}_{\text{upsample}} = \frac{|D_1| + |D_2|}{2|D_1|} \mathcal{L}_{D_1} + \frac{|D_1| + |D_2|}{2|D_2|} \mathcal{L}_{D_2},$$

   where we assume $D_1$ and $D_2$ are the two training datasets with different channels.

2. Concatenate the two datasets and draw a batch from the combined datasets. This approach simply minimizes the average loss

$$\mathcal{L}_{\text{average}} = \mathcal{L}_{D_1} + \mathcal{L}_{D_2}.$$

Our preliminary experiments indicate that the second method consistently outperformed the first. For instance, in JUMP-CP when training with 25% 8-channel data, ChannelViT-S/16 achieves 57.97% when training with $\mathcal{L}_{\text{average}}$ but only reachs 45.52% when training with $\mathcal{L}_{\text{upsample}}$. Similarly, ViT-S/16 achieves 50.29% when training with $\mathcal{L}_{\text{average}}$ but only scores 42.58% when training with $\mathcal{L}_{\text{upsample}}$. We hypothesize that models exhibit overfitting when trained using the upsampling loss. Therefore, we report the numbers for the normal average loss $\mathcal{L}_{\text{average}}$ in Table 3 and Figure 5.

### A.4   EVALUATION ACROSS ALL CHANNEL COMBINATIONS

To assess the channel robustness of the trained models, we enumerate all possible channel combinations and report the corresponding accuracy for each.

For instance, in Table 2, we have considered two training scenarios: the top section pertains to training on 5 fluorescence channels, while the bottom section pertains to training on all 8 channels. For the top section, we can evaluate the models for all subsets of the 5 fluorescence channels. This includes

- Combinations with 5 channels: there is only one $C_5^5 = 1$ combination;
- Combinations with 4 channels: there are $C_5^4 = 5$ combinations;
- Combinations with 3 channels: there are $C_5^3 = 10$ combinations;
- Combinations with 2 channels: there are $C_5^2 = 10$ combinations;
- Combinations with 1 channels: there are $C_5^1 = 5$ combinations.

Consequently, we evaluate a total of $1+5+10+10+5=31$ channel combinations. Given a specific channel combination, we mask out the testing images accordingly (as described in Appendix A.1) and compute the corresponding testing accuracy. We then report the average accuracy over combinations that have the same number of channels. As one might intuitively expect, models tend to perform better when provided with more channels.

## A.5 DATASET DETAILS

In this section, we provide a detailed description of our datasets and their corresponding input channels.

**JUMP-CP, 160-way classification**   We use the processed version of JUMP-CP released by Bao & Karaletsos (2023)[§]. Each image consists of a single masked cell and includes five fluorescence channels: AGP, DNA, ER, Mito, RNA, as well as three brightfield channels: HighZBF (Brightfield-1), LowZBF (Brightfield-2), and Brightfield (Brightfield-3). Each cell has been perturbed by a chemical compound, and the goal is to identify the gene target of the chemical perturbation.

**So2Sat, 17-way classification**   We use the processed version So2Sat released by the original authors Zhu et al. (2020a)[¶]. Each image patch consists of 8 channels from Sentinel-1:

1. the real part of the unfiltered VH channel;
2. the imaginary part of the unfiltered VH channel;
3. the real part of the unfiltered VV channel;
4. the imaginary part of the unfiltered VV channel;
5. the intensity of the refined LEE filtered VH channel;
6. the intensity of the refined LEE filtered VV channel;
7. the real part of the refined LEE filtered covariance matrix off-diagonal element;
8. the imaginary part of the refined LEE filtered covariance matrix off-diagonal element.

and 10 channels from Sentinel-2: Band B2, Band B3, Band B4, Band B5, Band B6, Band B7, Band B8, Band B8a, Band B11 and Band B12. The task is to predict the climate zone for each respective image patch, with a total of 17 distinct climate zones being represented.

## B ADDITIONAL BASELINES

### B.1 BASELINE: CONCATENATING FEATURES FROM MULTIPLE SINGLE-CHANNEL VITS

Hussein et al. (2022) utilized ViTs for epileptic seizure predictions, proposing a method to train multiple ViTs, one for each input channel. The final image representation is derived by aggregating the output `CLS` tokens across all single-channel ViTs. An MLP is then attached to these aggregated features to predict the image label. In this section, we implement this baseline based on the paper, termed MultiViT, and evaluate its performance both with and without HCS.

Table 5 presents our results on JUMP-CP when training using all eight channels. Without HCS, MultiViT underperforms compared to ViT when evaluated on all channels, despite having eight times more parameters. This underscores the importance of parameter sharing across different channels to combat overfitting. However, when testing on a subset of channels, MultiViT outperforms ViT, as each ViT operates on a single channel, thereby improving robustness to changes in the input channels. Interestingly, MultiViT does not perform well with HCS. While the accuracy improves when testing on a subset of channels, the accuracy significantly decreases (from 49.06 to 30.25) when using all eight channels. We hypothesize that this is due to the channel-wise feature aggregation being performed after the single-channel ViTs, preventing the model from conditioning the representation based on the input channel availability.

---

[§]https://github.com/insitro/ContextViT
[¶]https://github.com/zhu-xlab/So2Sat-LCZ42

Table 5: 160-way test accuracy of MultiViT (Hussein et al., 2022) on JUMP-CP. All models are based on the ViT-S/16 backbone and are trained on all 8 channels. During testing, all possible channel combinations are evaluated and we report the mean accuracies for combinations with the same number of channels. MultiViT learns a separate ViT per channel and aggregates their output `CLS` tokens together to form the overall image representation. Since the ViT encoder is separate for each channel, it offers better channel robustness than the vanilla ViT. However, if we focus on the performance using all channels, it actually leads to worse performance than the vanilla ViT-S/16, highlighting the importance of parameter tying on the application of cell imaging.

| | | ViT S/16 | MultiViT S/16 | ChannelViT S/16 | ViT S/16 | MultiViT S/16 | ChannelViT S/16 |
|---|---|---|---|---|---|---|---|
| Use hierarchical channel sampling? | | ✗ | ✗ | ✗ | ✓ | ✓ | ✓ |
| #channels for testing | 8 channels | 52.06 | 49.06 | 66.22 | 56.87 | 30.25 | **68.09** |
| | 7 channels | 5.91 | 34.10 | 41.03 | 49.35 | 29.04 | **61.02** |
| | 6 channels | 1.81 | 23.77 | 24.57 | 42.38 | 27.44 | **53.45** |
| | 5 channels | 2.46 | 17.09 | 14.20 | 35.78 | 25.69 | **45.50** |
| | 4 channels | 2.38 | 12.98 | 8.56 | 29.84 | 23.96 | **37.37** |
| | 3 channels | 2.70 | 10.58 | 5.65 | 24.94 | 22.34 | **29.68** |
| | 2 channels | 2.63 | 9.61 | 3.24 | 21.54 | 20.89 | **23.77** |
| | 1 channel | 3.00 | 7.97 | 2.08 | 19.92 | 19.85 | **20.84** |

Table 6: 160-way test accuracy of FAN (Zhou et al., 2022) on JUMP-CP. All models are trained on all 8 channels. During testing, we evaluated all possible channel combinations and reported the mean accuracies for combinations with the same number of channels. FAN incorporates a channel-wise self-attention mechanism following the standard location-wise self-attention in the transformer encoder. This enhances the model's ability to reason across both input and hidden channels, outperforming the ViT baseline. However, it remains sensitive to the availability of input channels.

| #channels for testing | Without HCS | | | | With HCS | | | |
|---|---|---|---|---|---|---|---|---|
| | ViT S/16 | FAN S/16 (conv patch) | FAN S/16 (linear patch) | ChannelViT S/16 | ViT S/16 | FAN S/16 (conv patch) | FAN S/16 (linear patch) | ChannelViT S/16 |
| 8 | 52.06 | 65.13 | 65.42 | *66.22* | 56.87 | 3.49 | 20.31 | **68.09** |
| 7 | 5.91 | 1.24 | 3.63 | *41.03* | 49.35 | 3.88 | 20.52 | **61.02** |
| 6 | 1.81 | 0.64 | 4.82 | *24.57* | 42.38 | 3.96 | 17.46 | **53.45** |
| 5 | 2.46 | 2.11 | 6.62 | *14.20* | 35.78 | 3.15 | 15.17 | **45.50** |
| 4 | 2.38 | 3.80 | 6.68 | *8.56* | 29.84 | 3.92 | 11.74 | **37.37** |
| 3 | 2.70 | 5.03 | *6.03* | 5.65 | 24.94 | 4.54 | 9.42 | **29.68** |
| 2 | 2.63 | 4.36 | *5.97* | 3.24 | 21.54 | 2.21 | 6.65 | **23.77** |
| 1 | *3.00* | 2.68 | 2.92 | 2.08 | 19.92 | 2.90 | 2.52 | **20.84** |

We find that ChannelViT significantly outperforms MultiViT. There are three key differences between the two models:

1. ChannelViT learns a single ViT across all channels, rather than one ViT for each channel;

2. ChannelViT is aware of the input channel availability at the input patch sequence, while the single-channel ViTs in MultiViT operate independently;

3. ChannelViT allows cross-channel cross-location attention, while MultiViT only permits cross-location attention.

## B.2 BASELINE: FULLY ATTENTIONAL NETWORKS (FANS)

Zhou et al. (2022) introduced a family of Fully Attentional Networks (FANs) that combine channel-wise attention with the MLP in a transformer encoder layer. Notably, the channels in this context

Table 7: 160-way test accuracy of ResNet50 and ResNet152 (Zhou et al., 2022) on JUMP-CP. All models are trained on all 8 channels. ResNets are trained without HCS. ViT and ChannelViT are trained with HCS. During testing, we evaluated all possible channel combinations and reported the mean accuracies for combinations with the same number of channels.

| Model | ResNet50 | ResNet152 | ViT-S/8 | ChannelViT-S/8 |
|---|---|---|---|---|
| #parameters | 25M | 60M | 22M | 22M |
| Use hierarchical channel sampling? | ✗ | ✗ | ✓ | ✓ |
| 8 Channels | 65.96 | 66.54 | 66.44 | 74.77 |
| 7 Channels | 2.39 | 3.05 | 59.01 | 68.42 |
| 6 Channels | 2.22 | 4.29 | 51.29 | 61.26 |
| 5 Channels | 1.57 | 5.35 | 43.39 | 53.05 |
| 4 Channels | 1.19 | 5.91 | 35.60 | 43.87 |
| 3 Channels | 0.78 | 5.69 | 28.59 | 34.19 |
| 2 Channels | 0.56 | 4.29 | 23.32 | 25.73 |
| 1 Channels | 0.51 | 2.66 | 20.41 | 21.20 |

extend beyond the input channels. FANs aggregate feature channels with high correlation values across the transformer encoder layers and isolate outlier features with low correlation values.

We adopted the implementation provided at `https://github.com/NVlabs/FAN/blob/master/models/fan.py` and evaluated the FAN small with a patch size of 16 by 16. It's worth noting that FAN, by default, employs four stacks of 3 by 3 convolution layers (each followed by GELU activations) to construct the input patch tokens, whereas ViT and ChannelViT use a single linear layer over the 16 by 16 input patches. We refer to this FAN baseline as FAN S/16 (conv patch). We also experimented with replacing these convolution layers with the same linear projection used in the regular ViT, terming this modified version of FAN as FAN S/16 (linear patch).

Table 6 presents our results on FANs. Without HCS, the default FAN-S/16 (conv patch) significantly outperforms ViT (65.13 vs 52.06), demonstrating the effectiveness of cross-channel attention. However, it still falls short of ChannelViT (65.13 vs. 66.22). Furthermore, when evaluated using a subset of channels at test time, its performance significantly declines (1.24 vs. 41.03 on 7 channels). Interestingly, we observed that the FAN with a linear patch embedding layer performs slightly better than the default FAN with convolution patch embeddings.

We also investigated training FANs with HCS. We discovered that FAN with convolution patch embeddings struggled to learn a meaningful classifier. Replacing the convolution layers with a simple linear transformation improved the performance, and we observed that when trained with HCS, FAN-S/16 (linear patch) outperforms its counterpart without HCS when evaluated on a subset of channels. However, the performance is still significantly lower than the regular ViT-S/16. We hypothesize that since FANs explicitly leverage the correlation between different hidden channels to build its representations, it becomes more sensitive to channel perturbations at test time.

In conclusion, we highlight the key differences between ChannelViT and FANs:

1. ChannelViT performs cross-channel and cross-location attention jointly, meaning that each patch token can attend to a different channel at a different location.

2. ChannelViT maintains the distinction of different input channels throughout the transformer encoder and tie the transformer encoder across channels, which we argue enhances robustness to channel changes.

### B.3 BASELINE: CONVOLUTIONAL NEURAL NETWORKS (CNNS)

In this section, we further compare the Vision Transformer (ViT) and ChannelViT with conventional Convolutional Neural Networks (CNNs). We use the widely-adopted ResNet-50 and ResNet-152 as our baseline models, as described by He et al. (2016). We present the number of parameters for each model and their corresponding performance on the JUMPCP dataset in Table7.

Initially, we trained the ResNet baselines using Hierarchical Channel Sampling (HCS), but this approach led to training instability, with the top-1 accuracies of both models converging to approximately 5% by the end of training. Without HCS, ResNet-50 and ResNet-152 exhibit performance comparable to the ViT-S/8 baseline. Despite having three times more parameters, ResNet-152 achieves only a slight improvement over ResNet-50. When compared with ChannelViT-S/8, there still remains a significant performance gap.

We hypothesize that the parameter sharing within ChannelViT enables the efficient and robust construction of channel-invariant filters. Conversely, the explicit cross-channel attention in ChannelViT effectively facilitates the model's ability to infer relationships across related channels.

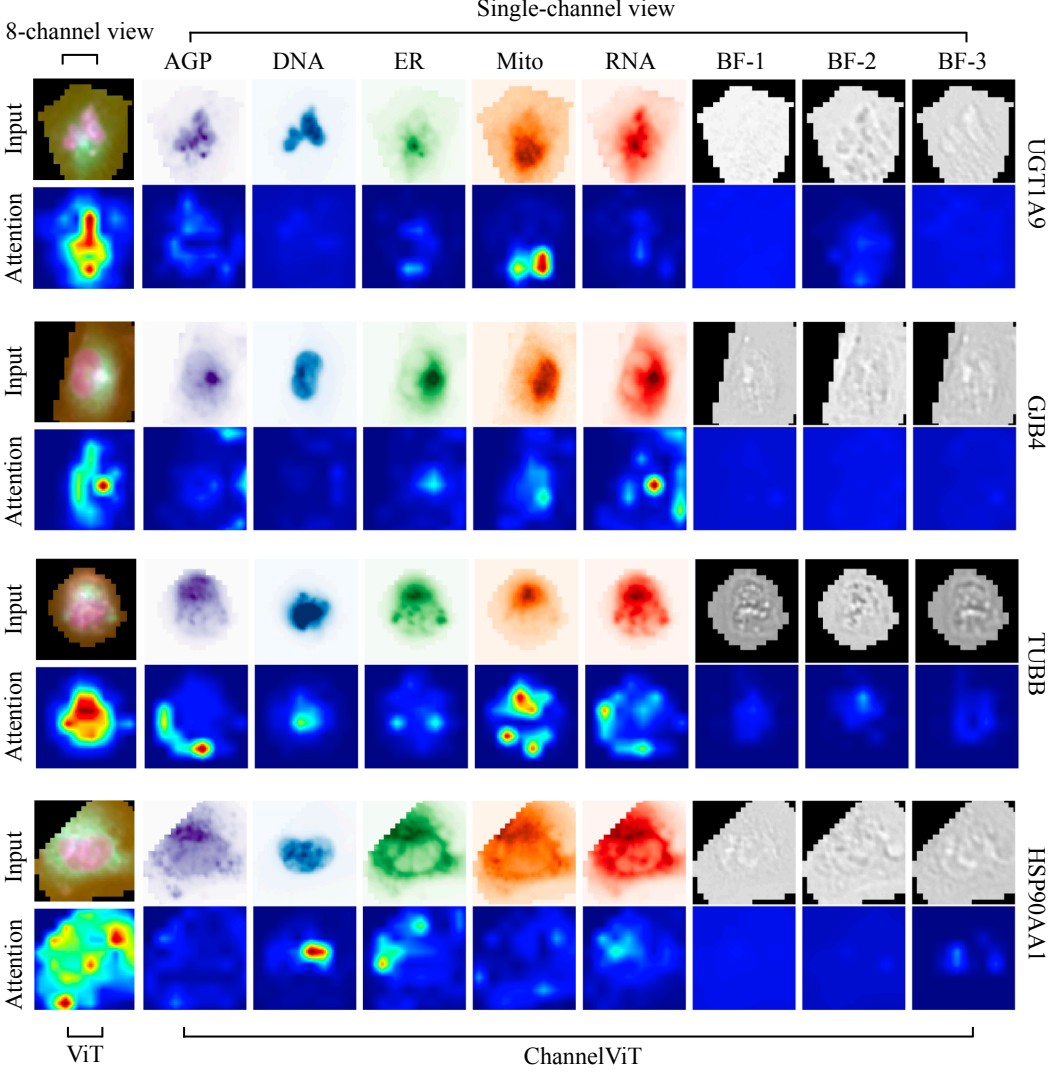

Figure 6: Extra visualizations of the relevance heatmaps for both ViT-S/8 (8-channel view) and ChannelViT-S/8 (single-channel view). Both models are trained on JUMP-CP using HCS across all 8 channels. ChannelViT offers interpretability by highlighting the contributions made by each individual channel.

Table 8: Time efficiency of different models on the JUMP-CP dataset. All models are trained and evaluated using all eight channels. We use a GPU cluster equipped with eight A100 GPUs for the evaluation. Section B.3 provides a more detailed analysis for ResNet models.

| Model | #parameters | Training time | Inference time | 8-channel accuracy |
|---|---|---|---|---|
| ResNet50 | 25M | 3.9 hours | 65.8 sec | 65.96 |
| ResNet152 | 60M | 4.4 hours | 81.8 sec | 66.54 |
| ViT-S/16 | 22M | **2.8** hours | **54.5** sec | 56.87 |
| ChannelViT-S/16 w/o HCS | 22M | 12.1 hours | 91.0 sec | 66.22 |
| ChannelViT-S/16 w/ HCS | 22M | 10.2 hours | 90.7 sec | **68.09** |

Table 9: ChannelViT-S/16: tied image filter vs. untied image filter. Both models are trained on the five fluorescence channels in JUMP-CP, with HCS applied during training. Tying the image filter weights across channels enhances both the performance and robustness.

| Tied linear projection weights? | | ✗ | ✓ |
|---|---|---|---|
| #channels for testing | 5 channels | 54.78 | 56.78 |
| | 4 channels | 43.88 | 45.94 |
| | 3 channels | 33.67 | 35.45 |
| | 2 channels | 25.57 | 26.57 |
| | 1 channel | 21.07 | 21.43 |

# C  ADDITIONAL ANALYSIS

## C.1  RUNNING TIME ANALYSIS

Our proposed ChannelViT model, which expands the channel dimension into the sequence length dimension, introduces an inherent increase in computational cost. As shown in Table 8, the training duration for the ChannelViT-S/16 model on the JUMP-CP dataset, utilizing all eight channels, is significantly longer without the application of Hierarchical Channel Sampling (HCS). However, the integration of HCS results in a 15% reduction in training time, decreasing from 12 hours and 6 minutes to 10 hours and 17 minutes. This demonstrates that HCS not only bolsters the model's robustness but also markedly enhances training efficiency.

In terms of inference cost, ChannelViT exhibits a 1.6 times increase in processing time compared to its ViT counterpart, yet it achieves an 11.22% higher accuracy (on an absolute scale). When measured against the better performing ResNet-152 baseline, Channel ViT's inference time is only 1.1 times longer.

In this paper, we have explored the ChannelViT utilizing the standard quadratic attention mechanism. Looking ahead, it would be intriguing to investigate the integration of ChannelViT with more efficient algorithms, such as Linformer Wang et al. (2020) and LongNet Ding et al. (2023), which scale linearly with sequence length. Such combinations could potentially yield further improvements in both performance and computational efficiency.

## C.2  ATTENTION VISUALIZATION FOR IMAGENET

Figure 7 illustrates the attention heatmaps for ViT and ChannelViT models trained on the ImageNet dataset. For each image, we generate the rolled out attention scores for two distinct classes—espresso and wine for the top image, and elephant and zebra for the bottom image—following Chefer et al. (2021). We observe that ChannelViT precisely focuses its attention on the relevant channels, such as the red channel when predicting red wine. In scenarios where the contrast pattern, such as black and white for a zebra, is distributed across all channels, ChannelViT effectively utilizes all channels to inform its prediction.

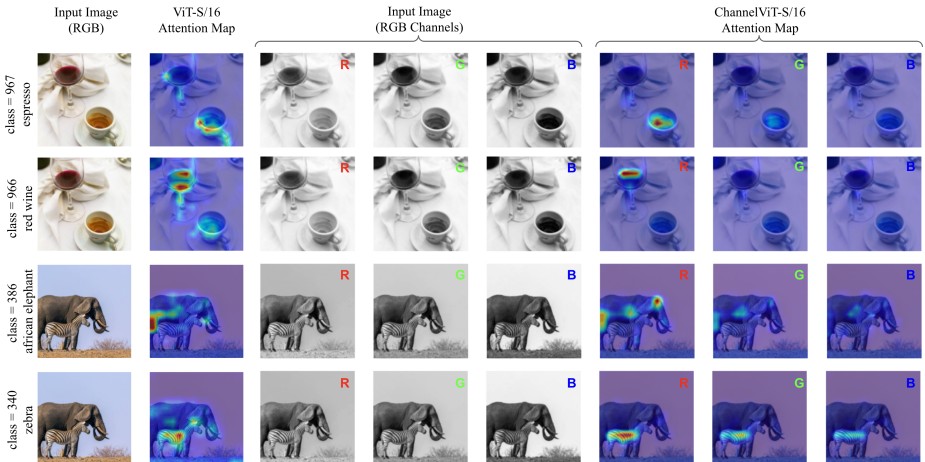

Figure 7: Relevance visualizations for ViT-S/16 and ChannelViT-S/16 trained on ImageNet. For each image, we generate the relevance heatmap for two distinct classes (espresso and wine for the top image, elephant and zebra for the bottom image) using the methodology described in Chefer et al. (2021). It's observed that ChannelViT precisely allocates its attention to the relevant channel (red channel for predicting red wine). In the case of predicting a zebra, where the black and white contrast pattern is present across all channels, ChannelViT utilizes all channels for its prediction.

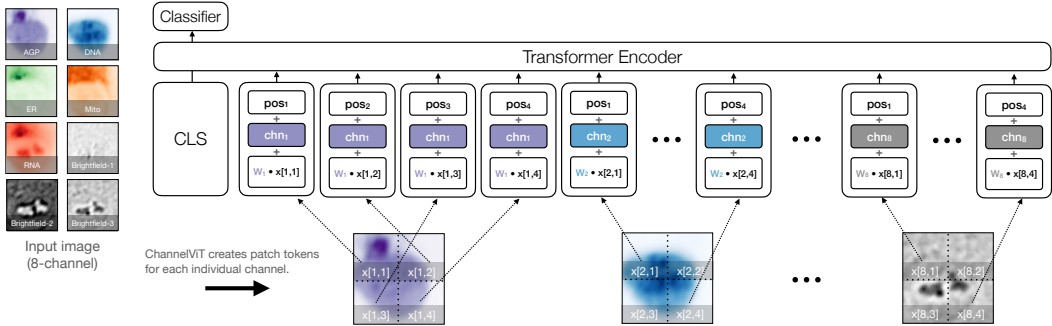

Figure 8: Illustration of ChannelViT with untied image filters. Each channel is assigned its unqiue linear projection weight, denoted as $W_c$. Contrarily, in Figure 1, all channel share a common image filter, represented by $W$.

## C.3 ABLATION: TIED VS. UNTIED IMAGE FILTERS FOR CHANNELVIT

In the main paper, we introduced ChannelViT with a linear projection layer tied across various channels. This section delves into the exploration of flexible weights for each channel (Figure 8). The input sequence to the Transformer encoder can be represented as follows:

$$\begin{bmatrix} \text{CLS}, & \text{pos}_1 + \text{chn}_1 + W_1 x[c_1, p_1], & \dots, & \text{pos}_N + \text{chn}_1 + W_1 x[c_1, p_N], \\ \dots, & \text{pos}_1 + \text{chn}_C + W_C x[c_C, p_1], & \dots, & \text{pos}_N + \text{chn}_C + W_C x[c_C, p_N] \end{bmatrix},$$

where $W_1, \dots, W_C$ denote the linear transformations associated with the input channels. Table 9 showcases our findings on JUMP-CP. It is observed that ChannelViT, when trained with tied image filter weights, consistently outperforms its untied counterpart. We hypothesize that the first layer filters are generally shareable across channels, and tying the parameters can prevent overfitting, thereby enhancing the model's robustness.

## C.4 ABLATION: SHARED VS. UNSHARED CHANNEL EMBEDDINGS

In this section, we conduct an ablation study to investigate the impact of channel embeddings on the performance of ChannelViT models. Specifically, we consider the following simplification of

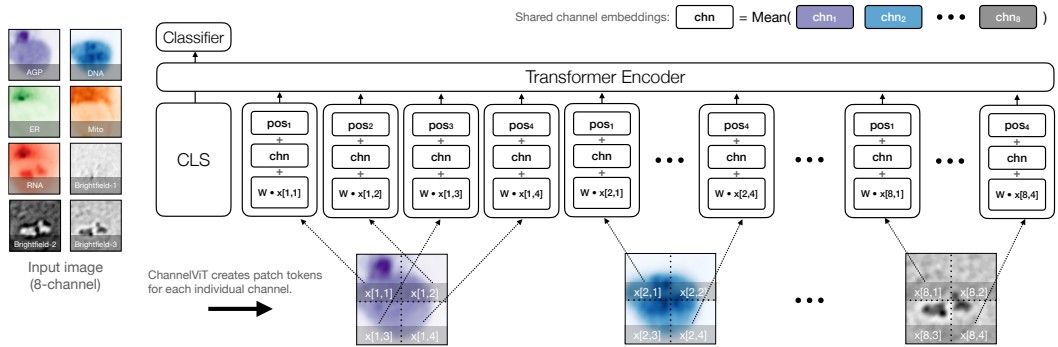

Figure 9: Illustration of ChannelViT with shared channel embeddings. We investigate the impace of channel embeddings on the performance of ChannelViT. Specifically, we set replaced each channel embedding by the mean channel embeddings across all channels. The resulting performance is presnted in Table 10.

Table 10: Ablation study on the channel embeddings of channelViT. The ChannelViT models, trained on JUMP-CP and So2Sat with hierarchical channel sampling across all channels, are utilized for this ablation study. For the top row (✓), the mean channel embeddings across all channels are computed and used to replace the channel embedding for each individual channel (see Figure 9). Observations indicate that the preservation of original channel embeddings is critical for both datasets. Notably, ChannelViT exhibits a higher sensitivity to modifications in channel embedding on JUMP-CP as compared to So2Sat.

| Shared channel embedding? | ChannelViT-S/16 on JUMP-CP | | ChannelViT-S/16 on So2Sat | |
| --- | --- | --- | --- | --- |
| | fluorescence (5 channels) | fluorescence & brightfield (8 channels) | Sentinel-1 (8 channels) | Sentinel-1 & -2 (18 channels) |
| ✓ | 1.26 | 2.49 | 10.44 | 52.13 |
| ✗ | 57.60 | 68.09 | 47.39 | 63.01 |

ChannelViT where we have a shared channel embedding across all channels:

$$\big[\text{CLS}, \quad \text{pos}_1 + \text{chn} + Wx[c_1, p_1], \quad \ldots, \quad \text{pos}_N + \text{chn} + Wx[c_1, p_N],$$
$$\ldots, \quad \text{pos}_1 + \text{chn} + Wx[c_C, p_1], \quad \ldots, \quad \text{pos}_N + \text{chn} + Wx[c_C, p_N]\big].$$

We consider ChannelViTs trained on both JUMP-CP and So2Sat. A natural way to define chn is to set it as the mean embeddings of the learned channel embeddings:

$$\text{chn} = \frac{1}{C} \sum_c \text{chn}_c.$$

We present our ablation study in Table 10. We observe that this modification significantly harms the performance, underscoring the importance of maintaining the original channel embeddings. Interestingly, the ChannelViT model demonstrates a higher degree of sensitivity to alterations in channel embedding on the JUMP-CP dataset as compared to the So2Sat dataset. This suggests that the specific characteristics of the dataset can influence the model's reliance on channel embeddings.

### C.5 INVESTIGATION: DO WE NEED A SEPARATE CLASSIFIER FOR EACH CHANNEL COMBINATION?

The application of hierarchical channel sampling results in the model receiving a variety of input channel combinations, leading to significant changes in the input distribution. This prompts an investigation into whether it's necessary to further condition the final classifier based on the sampled channel combinations. Table 11 presents our ablation analysis, where we consider three methods for incorporating the information of the input channels into the final classifier:

1. The first baseline involves learning a separate linear classifier on top of the ViT embeddings for each channel combination.

Table 11: Assessing the necessity of conditioning the classifier on input channel combinations during hierarchical channel sampling. The backbone models, ChannelViT-S/16, are trained and tested on 8 channels. Our findings suggest that the simplest shared linear classifier (bottom) delivers superior results, eliminating the need for extra conditioning. We hypothesize that the utilization of a shared linear classifier contributes to the regularization of the model's internal representation, thereby enhancing its robustness across channels.

| Additional features besides last-layer '[CLS]' | Classifier on top of the features | Classifier shared across channel combinations? | Accuracy |
|---|---|---|---|
| *Informing the final classifier of the sampled channel combination* | | | |
| None | Linear | ✗ | 26.56 |
| Embeddings for each channel comb. | MLP | ✓ | 61.98 |
| One-hot encoding of each channel comb. | MLP | ✓ | 66.86 |
| *ChannelViT* | | | |
| None | Linear | ✓ | **68.09** |

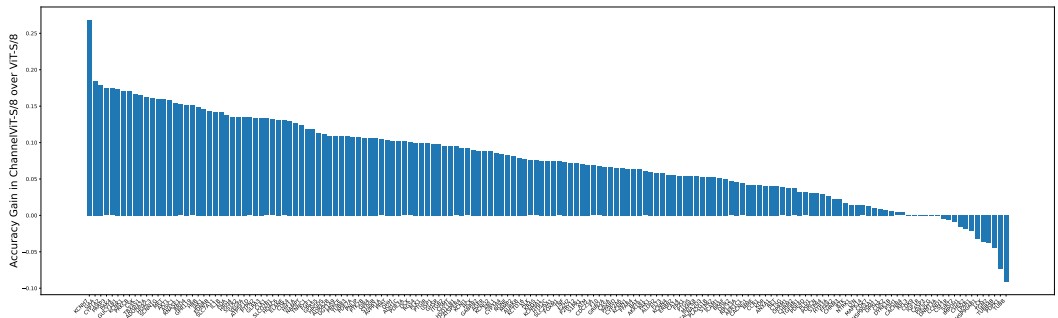

Figure 10: Accuracy gain of ChannelViT-S/8 over ViT-S/8 over all cell labels (gene targets) on JUMP-CP. Both models are trained using HCS over all 8 channels.

2. The second baseline learns an embedding vector for each channel and constructs the representation for the sampled channel combination by summing up all the embeddings for the selected channels. This representation is then concatenated with the ViT representation and fed to a shared MLP with one hidden layer.

3. The third method is similar to the second baseline, but uses one-hot encoding as the representation for the sampled channel combination.

Our observations indicate that all three methods underperform when compared to the basic ChannelViT, which uses a shared linear classifier across all channel combinations. We hypothesize that the shared linear classifier regularizes the ViT to embed inputs with different channel combinations into the same space. This bias appears to enhance robustness and performance.

## C.6   Breaking down the performance gain on JUMP-CP for each gene target

In Figure 10, we delve into a comparative analysis of the performance between ChannelViT-S/8 and ViT-S/8 across each cell label (gene target). Our figure reveals that ChannelViT surpasses ViT in 90% of the gene targets, while underperforming in the remaining targets. It's important to note that the gain is computed from a 160-way classification task, where the models are trained to optimize the average loss across all gene targets. If we reframe the problem using a multi-task learning objective, the distribution of gains per gene could potentially differ, and we expect the improvements of ChannelViT to be more consistent.

## C.7   Backbone: Small vs. Base vs. Large

In the main body of the paper, we explored the ViT and ChannelViT across different resolutions (16x16 and 8x8). To provide a comprehensive analysis, we extend our investigation to include

Table 12: Configurations of ViT and ChannelViT with different backbone sizes.

| size | embed dim | depth | num heads | MLP hidden dim |
|------|-----------|-------|-----------|----------------|
| Small | 384 | 12 | 6 | 1536 |
| Base | 768 | 12 | 12 | 3072 |
| Large | 1024 | 24 | 16 | 4096 |

Table 13: Test accuracy of 160-way perturbed gene prediction on JUMP-CP. All models are trained on all 8 channels, which includes 3 additional brightfield channels. During testing, all possible channel combinations are evaluated and we report the mean accuracies for combinations with the same number of channels. Please refer to Table 12 for detailed model configurations.

| | | ViT (trained with HCS) | | | ChannelViT (trained with HCS) | | |
|---|---|---|---|---|---|---|---|
| | | Small/16 | Base/16 | Large/16 | Small/16 | Base/16 | Large/16 |
| #channels for testing | 8 channels | 56.87 | 57.85 | 57.96 | 68.09 | 68.53 | 68.87 |
| | 7 channels | 49.35 | 50.35 | 50.93 | 61.02 | 61.56 | 62.01 |
| | 6 channels | 42.38 | 43.98 | 43.98 | 53.45 | 53.53 | 53.59 |
| | 5 channels | 35.78 | 37.26 | 37.26 | 45.50 | 45.96 | 46.54 |
| | 4 channels | 29.84 | 30.82 | 30.82 | 37.37 | 37.90 | 38.09 |
| | 3 channels | 24.94 | 25.37 | 25.37 | 29.68 | 29.96 | 30.06 |
| | 2 channels | 21.54 | 21.73 | 21.73 | 23.77 | 23.78 | 23.62 |
| | 1 channel | 19.92 | 20.04 | 20.04 | 20.84 | 21.61 | 21.06 |

various backbone sizes. Adhering to the conventions established by Dosovitskiy et al., we evaluated the performance of the ViT and ChannelViT models with small, base, and large backbones. The specific configurations of these models are detailed in Table 12.

Performance metrics for the different model sizes are presented in Table 13. We note a trend of incremental performance improvements in both ViT and ChannelViT as the number of parameters increases. Concurrently, the performance disparity between ViT and ChannelViT remains consistent and significant.

### C.8   Performance variations across different channel combinations

Table 14 presents an analysis of the standard deviation for both ViT and ChannelViT on the JUMP-CP dataset, considering all channel combinations. We report the mean accuracies for groups categorized by the same amount of channels. It is important to note that, despite maintaining a constant number of channels, the informational content of different channel combinations can differ markedly, which is reflected in the substantial standard deviations observed in the table.

To further dissect and comprehend this variance, we examined the performance gains of ChannelViT over ViT for *each individual channel combination*. The mean improvements, along with their standard deviations, are presented in Table 15. Our analysis substantiates that the performance enhancements attributed to ChannelViT are not only consistent across various combinations but also notably significant.

## D   Camelyon17-WILDS: Medical Imaging for Histopathology

In this section, we introduce another dataset, Camelyon17-WILDS, which was not included in the main paper due to space limitations.

### D.1   Dataset

The Camelyon17-WILDS dataset encompasses 455k labeled images from five hospitals. The task involves predicting the presence of tumor tissue in the central region of an image. Although the dataset employs standard RGB channels, these are derived from the hematoxylin and eosin stain-

Table 14: Analyzing the standard deviation of 160-way perturbed gene prediction on JUMP-CP. We evaluated all possible channel combinations during testing. We report the mean accuracies for groups of combinations that include the same number of channels. However, even when the number of channels is held constant, the amount of information conveyed by different channel combinations can vary significantly, leading to a large standard deviation in our results. To isolate and understand this variance, we present the mean and standard deviation of the **performance differences** between ChannelViT and ViT for each channel combination in Table 15.

| | | ViT-S/16 | ChannelViT-S/16 | ViT-S/8 | ChannelViT-S/8 |
|---|---|---|---|---|---|
| | Use hierarchical channel sampling? | ✓ | ✓ | ✓ | ✓ |
| | *Training on all 8 channels (5 fluorescence channels & 3 brightfield channels)* | | | | |
| | 8 channels ($C_8^8 = 1$) | 56.87 | 68.09 | 66.44 | **74.77** |
| | 7 channels ($C_7^8 = 8$) | 49.35±9.38 | 61.02±9.78 | 59.01±10.07 | **68.42**±9.11 |
| #channels for testing | 6 channels ($C_6^8 = 28$) | 42.38±10.64 | 53.45±12.40 | 51.29±12.47 | **61.26**±11.91 |
| | 5 channels ($C_5^8 = 56$) | 35.78±10.18 | 45.50±13.23 | 43.39±12.89 | **53.05**±13.41 |
| | 4 channels ($C_4^8 = 70$) | 29.84±8.32 | 37.37±12.25 | 35.60±11.55 | **43.87**±13.36 |
| | 3 channels ($C_3^8 = 56$) | 24.94±5.43 | 29.68±9.22 | 28.59±8.38 | **34.19**±11.10 |
| | 2 channels ($C_2^8 = 28$) | 21.54±2.37 | 23.77±4.89 | 23.32±4.27 | **25.73**±6.57 |
| | 1 channel ($C_1^8 = 8$) | 19.92±0.51 | 20.84±1.64 | 20.41±1.26 | **21.20**±2.17 |

Table 15: Improvements of ChannelViT over its ViT counterpart for the JUMP-CP microscopy cell imaging benchmark. We evaluated all possible channel combinations during testing. For each specific combination, we calculated the performance gains of ChannelViT over ViT. These improvements are summarized as mean values with corresponding standard deviations (std) for groups categorized by the same number of channels. Despite the considerable variability inherent in different channel combinations, our findings reveal that the performance enhancements of ChannelViT are both consistent and substantial.

| | | ChannelViT-S/16 over ViT-S/16 | ChannelViT-S/16 over ViT-S/16 | ChannelViT-S/8 over ViT-S/8 |
|---|---|---|---|---|
| | Use hierarchical channel sampling? | ✗ | ✓ | ✓ |
| | *Training on all 8 channels (5 fluorescence channels & 3 brightfield channels)* | | | |
| | 8 channels ($C_8^8 = 1$) | 14.16 | 11.22 | 8.32 |
| | 7 channels ($C_7^8 = 8$) | 35.13 ± 18.37 | 11.67 ± 1.17 | 9.41 ± 1.80 |
| #channels for testing | 6 channels ($C_6^8 = 28$) | 22.76 ± 18.64 | 11.07 ± 2.22 | 9.96 ± 1.90 |
| | 5 channels ($C_5^8 = 56$) | 11.75 ± 11.33 | 9.72 ± 3.46 | 9.66 ± 2.30 |
| | 4 channels ($C_4^8 = 70$) | 6.18 ± 6.86 | 7.52 ± 4.19 | 8.27 ± 3.08 |
| | 3 channels ($C_3^8 = 56$) | 2.95 ± 5.27 | 4.74 ± 3.96 | 5.60 ± 3.58 |
| | 2 channels ($C_2^8 = 28$) | 0.61 ± 3.97 | 2.24 ± 2.62 | 2.41 ± 2.82 |
| | 1 channel ($C_1^8 = 8$) | −0.92 ± 7.49 | 0.93 ± 1.15 | 0.79 ± 0.95 |

ing procedure, which can vary across hospitals. We adopt the processed version from the WILDS benchmark[‖].

## D.2 RESULTS

Table 16 presents our results for Camelyon17, a medical imaging benchmark for histopathology. Given the smaller image size (96 by 96), we employ a patch size of 8 by 8 for the ViT backbone.

Starting with the standard ViT-S/8 (first column), we note that it achieves an accuracy of 99.14 for the in-distribution hospitals. With HCS, it also attains an accuracy of over 97 when using only two or one channels for predictions. However, when evaluated on out-of-distribution hospitals, its 3-channel accuracy drops to 83.02. This is not only lower than its in-distribution performance,

---

[‖] https://wilds.stanford.edu

Table 16: Test accuracy of binary cancer classification on Camelyon17-WILDS. We consider all channel combinations and report the mean accuracy over combinations with the same number of channels. All models are trained with HCS. We observe 1) tying the linear patch projection layer across channels improves out-of-distribution generalization; 2) ChannelViT outperforms ViT on out-of-distribution hospitals.

| | ViT-S/8 | ViT-S/8 | ViT-B/8 | ChannelViT-S/8 | ChannelViT-S/8 | ChannelViT-B/8 |
|---|---|---|---|---|---|---|
| Tied weights across channels? | ✗ | ✓ | ✓ | ✗ | ✓ | ✓ |
| *Evaluation on in-distribution hospitals* | | | | | | |
| 3 channels | **99.14** | 98.46 | 98.28 | 98.98 | 98.99 | 99.13 |
| 2 channels | 98.65 | 98.42 | 98.22 | 98.51 | **98.66** | 98.73 |
| 1 channel | 97.59 | **98.24** | 97.98 | 97.71 | 98.14 | 98.11 |
| *Evaluation on out-of-distribution hospitals* | | | | | | |
| 3 channels | 83.02 | 89.14 | 88.57 | 89.96 | **92.67** | 91.39 |
| 2 channels | 85.12 | **88.78** | 88.32 | 88.11 | 88.25 | 87.17 |
| 1 channel | 87.97 | 87.19 | 86.93 | 87.04 | **88.30** | 87.60 |

Table 17: Top-1 Accuracy on ImageNet Using DINO Pre-training with ViT and ChannelViT. We apply DINO pre-training with both ViT and ChannelViT on the ImageNet training data. Upon completion of the pre-training phase, we conduct the standard linear probing evaluation , and the resultant validation accuracy is reported. Hierarchical channel sampling is not used as we found that it introduces extra instability during the DINO pre-training phase. The findings indicate that 1) In comparison to supervised training, DINO inherently enhances the channel robustness for ViT; 2) ChannelViT consistently outperforms ViT in a significant manner across all evaluations.

| Backbone | Val Acc. on RGB | Val Acc. on R-only | Val Acc. on G-only | Val Acc. on B-only |
|---|---|---|---|---|
| *Models trained on three channels (RGB)* | | | | |
| Supervised ViT-S/16 | 71.49 | 29.39 | 33.79 | 21.18 |
| DINO + ViT-S/16 + LinearProb | 72.62 | 64.34 | 65.46 | 61.12 |
| DINO + ChannelViT-S/16 + LinearProb | 74.38 | 67.44 | 67.85 | 65.97 |
| *Expert DINO models pre-trained on only one channel* | | | | |
| DINO + ViT-S/16 (R-only) + LinearProb | — | 67.76 | — | — |
| DINO + ViT-S/16 (G-only) + LinearProb | — | — | 68.09 | — |
| DINO + ViT-S/16 (B-only) + LinearProb | — | — | — | 66.65 |

but also lower than the accuracy achieved when using only one channel for evaluation in the out-of-distribution hospitals (87.97). We hypothesize that this discrepancy is due to the *staining shift* across hospitals Gao et al. (2022). The mismatch in color distributions results in out-of-distribution inputs for the first linear patch embedding layer. To test this hypothesis, we experiment with tying the parameters across different channels for the first linear patch embedding layer. As seen in the second column, ViT-S/8 with tied weights, while performing slightly worse in the in-distribution hospitals, performs significantly better in the out-of-distribution setting. We also explore ViT-B/8 but found it exhibited overfitting.

By default, we share the first linear patch embedding layer across different channels for ChannelViT. On the out-of-distribution hospital, ChannelViT-S/8 significantly outperforms ViT-S/8 (92.67 vs. 89.14). We also observe that if we untie the weights for different channels in ChannelViT, the generalization performance degrades.

# E   SELF-SUPERVISED PRE-TRAINING WITH CHANNELVIT

This section delves into the integration of self-supervised learning with ChannelViT.

### E.1 DINO

We use the DINO algorithm (Caron et al., 2021) for self-supervised learning. It involves a self-distillation process where the student model, provided with local views of the input image, has to learn from the teacher model which has the global views of the same input image.

We follow most of the the configuration suggested by DINO repository[**]. Specifically, we pre-train DINO with ViT-S/16 and ChannelViT-S/16 for a total of 100 epochs on ImageNet with a batch size of 256. The AdamW optimizer (Loshchilov & Hutter, 2019) is employed, and the learning rate warm-up phase is set for the first 10 epochs. Given our batch size, the maximum learning rate is set to 0.0005, in line with recommendations from You et al. (2018). The learning rate is subsequently decayed using a cosine learning rate scheduler, with a target learning rate of $10^{-6}$. Weight decay is applied to all parameters, excluding the biases. The initial weight decay is set to 0.04 and is gradually increased to 0.4 using a cosine learning rate scheduler towards the end of training. The DINO projection head utilized has 65536 dimensions, and batch normalization is not employed in the projection head. The output temperature of the teacher network is initially set to 0.04 and is linearly increased to 0.07 within the first 30 epochs. The temperature is maintained at 0.07 for the remainder of the training. To enhance training stability, the parameters of the output layer are frozen during the first epoch.

### E.2 LINEAR PROBING

Upon the completion of the pre-training phase, the parameters of both ViT and ChannelViT are frozen. In alignment with the methodology proposed by Caron et al. (2021), the final four layers of the CLS representation are concatenated to represent the image. Subsequently, a linear classifier is trained on this image representation. The training of the linear classifier is conducted using SGD, with a learning rate of 0.005 and a momentum value of 0.9. The learning rate is decayed in accordance with a cosine annealing scheduler. We train the linear classifier for 100 epochs using the ImageNet training split. Once training is done, we report its Top-1 accuracy on the validation split.

### E.3 RESULTS

Table 17 showcases our results. It is noteworthy that hierarchical channel sampling is not used during DINO pre-training due to its potential to introduce additional instability to the self-distillation objective. However, we observe that DINO-pretrained ViT inherently provides superior channel robustness. Compared to the supervised ViT-S/16, it achieves 64.34 on the red-only evaluation, which is 34.95 better than its supervised version. Furthermore, the integration of DINO-pretraining with ChannelViT consistently enhances performance across all evaluations, bridging the gap towards the expert DINO model that is pre-trained on each individual channel.

---

[**]https://github.com/facebookresearch/dino

