# OpenReview forum: "Channel Vision Transformers: An Image Is Worth 1 x 16 x 16 Words"
_ICLR.cc/2024/Conference — ICLR 2024 poster_

### Official Review · Reviewer_rJx2 · 2023-10-31

**Soundness:** 4 excellent
**Presentation:** 3 good
**Contribution:** 3 good
**Rating:** 8
**Confidence:** 4

**Summary:**

The authors propose a new tokenization scheme for Vision Transformer (ViT), where an input image is split into patches not only in spatial dimensions but also in channel dimensions. This may be beneficial for data where channels carry different semantical information, thus having little correlation, and might not always be present altogether in the data. Additionally to the tokenization scheme, the authors propose Hierarchical Channel Sampling (HCS) to regularize training, adapting the model to imperfect inputs.

**Strengths:**

* The proposed method outperforms the original ViT model in almost all considered experiments.
* The proposed regularization training scheme proves its efficiency against input channel dropout even for the original ViT.
* It is claimed that it is possible to increase explainability through attention map analysis with the new scheme.
* The experimental base behind the work is diverse and extensive. Most authors’ statements are supported by an experiment or an ablation in the main section or the Appendix.
* The paper is easy to read and understand (illustrations, tables, etc.), and well-organized.

**Weaknesses:**

* The novelty of the work is limited, as it seems that the channel splitting proposed in this paper is similar to a multi-modal approach to multi-channel data that has been explored earlier (albeit not with images). Also, HCS can be viewed as a modification of input channel dropout that has also been known before.
* The gains from the proposed approach are limited to the data with many channels (e.g., some microscopy images) and seem less reasonable for other types. At the same time, the processing with this new approach takes significantly more time.

**Questions:**

1. I suggest revisiting the title. The inspiration for the title used in this paper has clearly come from the ViT’s “An image is worth 16x16 words”. But in the case of ViT, it was about patches of size 16x16 across all channels, i.e., "Cx16x16", while the proposed approach uses "1x16x16" "words", as the image is split into channels as well.
2. Please clarify why the proposed channel sampling is “hierarchical.”
3. It would be beneficial to review the notation used in the formulae, e.g. use AB instead of A\cdotB for matrix multiplication.

---

> ### Author Response · Authors · 2023-11-21
>
> We are grateful for your thorough review and constructive feedback, which have been very important in refining our manuscript.
> + **Novelty:** We wish to emphasize the novel aspects of our work in comparison to the existing literature:
>   + **Robustness of ViT to Missing Channels:** Prior research has explored various robustness measures for ViT models, such as resilience to spurious correlations and adversarial attacks. Our work pioneers the investigation into the channel robustness of ViTs, addressing realistic scenarios in multi-channel imaging applications where sensor discrepancies between training and testing are common.
>   + **ChannelViT for modeling multi-channel images:** ChannelViT is designed to create patch tokens from each channel, enabling effective cross-channel and positional reasoning. Unlike the main literature, we employ weight sharing across channels and leverage learnable channel embeddings to encapsulate channel-specific information. In Appendix B, we present additional comparisons with recent ViT variants for multi-channel imaging, where our simple ChannelViT demonstrates significant performance enhancements.
>   + **HCS as a Training Regularizer:** HCS not only improves training efficiency but also substantially bolsters model robustness when evaluated on varying channel combinations. It is true that it is largely inspired by channel dropout, and we have delineated the similarities and distinctions between HCS and channel dropout within the paper. Figure 3 showcases its superior efficacy, even when combined with a standard ViT.
> + **Limited to Data with Multiple Channels:** ChannelViT is indeed tailored for modeling cross-channel interactions, a critical yet previously neglected issue in multi-channel imaging. As shown in Table 2, the advantages of ChannelViT amplify with the diversity of input sources. In the paper we have also conducted an exhaustive evaluation of ChannelViT on conventional imaging datasets:
>   + On ImageNet, ChannelViT outperforms a standard ViT even when both are trained and evaluated on all channels. ChannelViT's robustness is evident when tested with missing channels.
>   + On Camelyon17-WILDS, ChannelViT's parameter sharing across channels results in enhanced generalization to new hospitals compared to regular ViT.
> + **Computational Cost:**
>   + A new section (Section 4.2) has been added to the manuscript to address computational efficiency, with an extensive runtime analysis in Table 8. Although training ChannelViT requires approximately 3.64 times more resources, the inference time is merely 1.6 times longer than that of ViT. We anticipate future integration of ChannelViT with more efficient attention mechanisms, such as Linformer or LongNet, will lead to further efficiency gains.
>   + It is also noteworthy that the additional training time is counterbalanced by eliminating the need to train separate expert models for each channel combination. ChannelViT's performance at processing inputs with varying channels translates to significant practical savings.
> + **Title Revision:** We really appreciate your careful consideration of the title's interpretation. The original ViT paper examined patch sizes of both 16x16 and 14x14, with the latter yielding superior performance. Given their use of a 224x224 training resolution, this equates to 16x16 patch tokens (224/14=16) for each input image. In our work, as we create image patches for each input channel, this results in Cx more patch tokens (words). We acknowledge the potential for confusion and are considering updating the title for better clarity.
> + **Hierarchical Channel Sampling:** In HCS, we first determine the number of channels to utilize (m). Subsequently, based on m, we select the specific channel combination. The term "hierarchical" is derived from the two-stage nature of the sampling process.
> + We have amended the notation in the formula as per your suggestion—thank you for bringing this to our attention.
>
> We hope that this rebuttal comprehensively addresses the concerns you have raised. If you have any other questions or concerns, please let us know. We thank you once again for your valuable time and input.

---

> > ### Comment · Reviewer_rJx2 · 2023-11-22
> > **Response to the Authors**
> >
> > Dear Authors,
> >
> > Thank you for your comments. I have read them.
> >
> > You have addressed many of my concerns, yet I remain slightly unconvinced that the procedure you refer to as HCS warrants such a complex name. Otherwise, I do not have other major questions.

---

### Official Review · Reviewer_EcAm · 2023-10-31

**Soundness:** 3 good
**Presentation:** 3 good
**Contribution:** 3 good
**Rating:** 8
**Confidence:** 5

**Summary:**

The paper presents an extension of the Vision Transformer (ViT) architecture for image analysis domains with multi-channel information. This includes microscopy (fluorescence), and satellite (spectral) imaging, among others. The method consists of decoupling channels from the input and passing them as different tokens. To this end, the authors designed a learnable channel embedding and a hierarchical sampling method. As a result, a ChannelViT model learns to associate tokens in the space and channel dimensions, and is robust to channel drops.

**Strengths:**

* Well motivated paper. The problem of processing multi-channel images is important and the paper presents arguments to design methods that deal with this information effectively.
* The method is simple and effective. Splitting images in a sequence of tokens is an intuitive approach and the results show its effectiveness.
* Beyond tokenizing, an important aspect of the method is sampling channels strategically to learn their associations.
* The analysis in various datasets, including ImageNet, JUMP-CP and So2Sat, adds to the evidence that the method works in practice.
* Various ablations and extensive details presented in the manuscript and the appendix.
* The evaluation includes quantitative and qualitative results that show the benefits of the proposed approach.

**Weaknesses:**

Major comments:
* Limited baselines reported. The paper only considers ViTs as a baseline, which is a natural comparison given that the proposed method is an extension of this architecture. However, comparison to other architectures, especially CNNs to solve the multi-channel problems should be reported. This is specially important to appreciate the relative performance compared to other solutions tested before in multi-channel images.
* Computational cost increases quadratically with the number of channels. This problem is not addressed or commented in the main manuscript. According to the reported times in the appendix, the training cost increases linearly with the number of channels. The computational cost should be analyzed and discussed in more detail. While the solution seems effective, it can be prohibitive in practice.
* In a similar note as the first point, other adaptations of ViTs have been reported for video analysis and 3D imaging, which also increase the image dimensions in a new axis (different than channels). Do any of these existing adaptations apply for channels?
* The prediction of treatment in the JUMP-CP dataset is not necessarily a biologically relevant task. Prior literature makes the distinction that this is a pretext task for learning representations (weakly supervised learning), and not the main goal of analyzing these images. This clarification should be mentioned and prior work should be cited. Ideally, a biologically relevant task should be reported to ensure that classification performance in this pretext task is not dominated by confounders or spurious correlations. If this is not possible, an alternative test or explanation should be provided.

Other comments:
* The results in Table 3 are based on a random selection of images with different numbers of channels (25%, etc). The reviewer assumes that the random partition is the same across experiments. The results should be repeated a few times with different partitions and standard deviations should be reported.

Minor comments:
* Figure 5 reports KRAS in the top row and KCNH76 in the bottom row, but the main text refers to them in the opposite order.
* Does figure 6 include error bars? Can you clarify how these were obtained?
* Minor typos: demonstrats, utiliz, involvs.

**Questions:**

* Can the authors report additional baselines and compare to other state-of-the-art methods in the same datasets that they evaluate? This could clarify the achievements of the method in context to prior art in these domains.
* Can the authors add computational analysis and comparisons to other methods for training / testing time of the methods? Even if the cost is a limitation, reporting and evaluating this aspect transparently can enhance the understanding of how useful this method can be in practice.

---

> ### Author Response · Authors · 2023-11-21
>
> We sincerely appreciate the detailed review and the constructive suggestions provided. Your feedback has been instrumental in enhancing the quality of our manuscript.
> + **Baselines:** We are grateful for the recommendation to include additional baselines. In response, we have expanded our paper to incorporate the following new results, detailed in Appendix B:
>   + **CNNs (Appendix B.3):** We conducted experiments with ResNet-50 and ResNet-152 on the JUMP-CP dataset. The outcomes are presented in Table 7. Our analysis reveals that both ResNet models exhibit performance comparable to ViT-S/8, with ResNet-152 slightly outperforming ResNet-50. However, when testing on different channel combinations, the ResNets' performance diminishes. Attempts to integrate ResNet with HCS during training resulted in instability, culminating in a final 8-channel accuracy of approximately 5%. Finally, the performance disparity between the new ResNet baselines and ChannelViT is consistent and substantial.
>   + **MultiViT (Appendix B.1):** Hussein et al. recently proposed the Multi-Channel ViT architecture for multi-channel EEG data analysis (https://pubmed.ncbi.nlm.nih.gov/35884859/). Their approach involves learning individual ViTs for each channel and fusing the outputs of their final layers for classification. Unlike MultiViTs, ChannelViT models cross-channel dependencies directly from the very first layer. Our comparative analysis of MultiViT, ViT, and ChannelViT, presented in Table 5, demonstrates that MultiViT, when trained without HCS, is more robust than ViT. Nonetheless, it consistently underperforms compared to ChannelViT. Incorporating HCS with MultiViT adversely affects its performance.
>   + **Fully Attentional Networks (Appendix B.2):** Zhou et al. introduced Fully Attentional Networks (FANs) that enhance a standard transformer encoder with channel-wise attention (arxiv.org/abs/2204.12451). Contrary to FANs, ChannelViT executes cross-channel and cross-location attention jointly, allowing each patch token to attend to different channels at distinct locations. FANs restrict attention to different channels within the same location. Utilizing the official implementation, we reported FANs' performance in Table 6. Training FANs without HCS significantly improves performance over a standard ViT (52.06 vs. 65.42). However, ChannelViT still slightly outperforms FANs (66.22 vs. 65.42). When trained with HCS, ChannelViT's performance markedly increases to 68.09, while FANs experience a significant performance decline (from 65.42 to 20.31).
>   + **Other baselines:** We acknowledge the reviewer's suggestions regarding other ViT variants for video analysis and 3D imaging. We argue that the data distribution in multi-channel imaging applications is fundamentally distinct from that of video or 3D data. Specifically, the images we considered are centered around the same object but captured with different sensors. We believe that while ChannelViT could potentially be adapted for sensor fusion beyond multi-channel imaging, such applications are beyond the scope of our current work.
> + **Computational Cost:**
>   + We have incorporated a new section (Section 4.2) to discuss computational efficiency, including a more comprehensive runtime analysis in Table 8. While training ChannelViT is approximately 3.64 times more resource-intensive, the inference time is only 1.6 times longer than that of ViT. We anticipate that integrating ChannelViT with more efficient attention mechanisms, such as Linformer or LongNet, could yield further improvements in efficiency.
>   + It is also noteworthy that the additional training time is counterbalanced by eliminating the need to train separate expert models for each channel combination. ChannelViT's performance at processing inputs with varying channels translates to significant practical savings.
> + **Task of Treatment Classification:** In line with the initial release of the JUMP-CP benchmark (https://www.biorxiv.org/content/10.1101/2022.01.05.475090v3), we have framed the task of perturbation detection (or treatment classification) as a means to assess representation models for cell imaging. We concur that the primary objective of cell image analysis extends beyond perturbation prediction to encompass other biologically pertinent tasks. We have clarified this point in our manuscript (Page 5, JUMP-CP section) and welcome the opportunity to reference any relevant prior work that you may suggest.
> + **Standard Deviation and Error Bars:**
>   + The results presented in Table 3 and Figure 5 were derived from multiple runs using different random seeds for both the partitions and the model initializations. We have now included a descriptive text to clarify these results.
>   + Table 14 details the standard deviations across different channel combinations. Additionally, Table 15 provides the mean and standard deviation of the improvements observed for each channel combination with ChannelViT over ViT.

---

> > ### Comment · Reviewer_EcAm · 2023-11-23
> > **Great job!**
> >
> > Dear authors,
> >
> > Thank you for the clarifications and for adding the new content and experiments to the paper.
> > I do not have additional questions or other concerns.

---

> ### Author Response · Authors · 2023-11-21
>
> [following previous comment]
> + **Typos:** We are grateful for your assistance in identifying typographical errors. These have been corrected in the revised version of our manuscript.
>
> We trust that this rebuttal comprehensively addresses the concerns you have raised. Should you have any further inquiries, please do not hesitate to contact us. We thank you once again for your valuable time and input.

---

### Official Review · Reviewer_ptju · 2023-11-01

**Soundness:** 3 good
**Presentation:** 3 good
**Contribution:** 2 fair
**Rating:** 5
**Confidence:** 4

**Summary:**

This work presents ChannelViT, a new ViT variant that: 1) learns a channel embedding for each channel, and 2) proposes a technique called Hierarchical Channel Sampling (HCS) for robustness to partial channels. Together, ChannelViT is reported to demonstrate better robustness to missing channels. Multiple experiments were performed, such as: 1) different model architectures (Channel variants of ViT-S/16 and ViT-S/8), 2) comparison with and without HCS, 3) comparisons across multiple benchmarks including microscopy imaging, 4) comparisons across varying number of channels available, 5) exploration into SSL compability in DINO, and others.

**Strengths:**

- ChannelViT with HCS proposes a simple but relatively intuitive extension of ViTs, which would have unique applications in multiplexed imaging in which not all "channels" (e.g. - fluorescent probes) are made available. Overall, this work presents a method that targets an important application and may enable significant biological / clinical findings in multiplexed imaging.
- The ablation experiments regarding assessment of partial channels, comparisons and baselines with training on single channels, and experimentation with DINO are thoughtful and well-organized. In particular, the assessment of partial channels and pretraining with DINO highlights its adaptability as a ViT in potentially replacing vanilla ViTs. Table 2 and Figure 4, which demonstrates the performances of Channel ViT (in comparison to vanilla ViT) on JUMP-CP with partial channels, demonstrates minor but consistent improvement.

**Weaknesses:**

- While the main application of ChannelViT is in targeting problems such as those in multiplexed imaging due to the challenge of generalizing encoders across datasets that would have the same set of probes, only one dataset explored in this work is related to multiplexed imaging. Experimentation on ImageNet, C17-WILDS, and others are informative and appreciated, are not as relevant to the ultimate application that ChannelViT serves. Though C17-WILDS is microscopy, hematoxylin and eosin (H&E) pathology is not a domain where only the hematoxylin or eosin stains are performed. Rather, it would be more interesting to explore this problem on other relevant multiplexed imaging benchmarks such as RXRX1 [1], RXRX1-WILDS [2], TissueNet [3], and others [4-6]. To this end, it would also be useful to discuss and compare ChannelViT in context with label-free approaches, which suggest channel synthesis as a paradigm for addressing missing channels.
- As a drop-in replacement for ViT in conventional natural image classification or medical imaging tasks, one concern I have (raised in the work) is lack of experimentation on model scale. The title of this work, "Channel Vision Transformers: An Image Is Worth C x 16 x 16 Words", is similar to a title of a related seminal work in ViTs [7], but does not evaluate and demonstrates the scale of Channel ViTs in the same manner in demonstrating its universality of overtaking  current state-of-the-art architectures. As ViT-Small architectures are not the most commonly-used model size for ViT, it would be informative to explore Channel ViTs at greater scale (such as ViT-B and ViT-L). An additional limitation connected to this is the efficiency of learning channel embedings, which may have been the reason for not developing larger ViT models. As shown in the DINO experimentation, even with HCS, training is approximately 3.64x more expensive than its baseline comparison, which may limit its usability.

References:
1. Haque, I., 2023. Rxrx1: A dataset for evaluating experimental batch correction methods. In Proceedings of the IEEE/CVF Conference on Computer Vision and Pattern Recognition (pp. 4284-4293).
2. Sypetkowski, M., Rezanejad, M., Saberian, S., Kraus, O., Urbanik, J., Taylor, J., Mabey, B., Victors, M., Yosinski, J., Sereshkeh, A.R. and Haque, I., 2023. Rxrx1: A dataset for evaluating experimental batch correction methods. In Proceedings of the IEEE/CVF Conference on Computer Vision and Pattern Recognition (pp. 4284-4293). (https://wilds.stanford.edu/datasets/)
3. Greenwald, N.F., Miller, G., Moen, E., Kong, A., Kagel, A., Dougherty, T., Fullaway, C.C., McIntosh, B.J., Leow, K.X., Schwartz, M.S. and Pavelchek, C., 2022. Whole-cell segmentation of tissue images with human-level performance using large-scale data annotation and deep learning. Nature biotechnology, 40(4), pp.555-565.
4. Bray, M.A., Gustafsdottir, S.M., Rohban, M.H., Singh, S., Ljosa, V., Sokolnicki, K.L., Bittker, J.A., Bodycombe, N.E., Dančík, V., Hasaka, T.P. and Hon, C.S., 2017. A dataset of images and morphological profiles of 30 000 small-molecule treatments using the Cell Painting assay. Gigascience, 6(12), p.giw014.
5. Cross-Zamirski, J.O., Mouchet, E., Williams, G., Schönlieb, C.B., Turkki, R. and Wang, Y., 2022. Label-free prediction of cell painting from brightfield images. Scientific reports, 12(1), p.10001.
6. Moshkov, N., Bornholdt, M., Benoit, S., Smith, M., McQuin, C., Goodman, A., Senft, R.A., Han, Y., Babadi, M., Horvath, P. and Cimini, B.A., 2022. Learning representations for image-based profiling of perturbations. Biorxiv, pp.2022-08.
7. Dosovitskiy, A., Beyer, L., Kolesnikov, A., Weissenborn, D., Zhai, X., Unterthiner, T., Dehghani, M., Minderer, M., Heigold, G., Gelly, S. and Uszkoreit, J., 2020. An image is worth 16x16 words: Transformers for image recognition at scale. arXiv preprint arXiv:2010.11929.

**Questions:**

Summarizing my above concerns, though this work would benefit from having additional experimentation related to its targeted application in multiplex imaging, such as:
- Additional evaluation on relevant datasets in multiplex imaging, such as RXRX1 and TissueNet
- Discussion and benchmark evaluation against label-free approaches in multiplex imaging
- Ablation experiments with ViT-B and ViT-L, with expanded discussion on the intended usage of ChannelViT w.r.t. efficiency

---

> ### Author Response · Authors · 2023-11-21
>
> Thank you for the detailed review and the constructive suggestions, which we have found to be very helpful.
> + **Scope of ChannelViT:** We acknowledge the references you provided, which pertain to cell imaging. It is important to clarify that ChannelViT is designed as a versatile image representation backbone suitable for a broad range of multi-channel imaging applications, not exclusively for cell imaging. To demonstrate this, we have selected four diverse datasets for a thorough evaluation of ChannelViT:
>   + **ImageNet:** As a foundational image representation benchmark, ImageNet is critical for assessing ChannelViT's performance in standard computer vision tasks. Our results on ImageNet demonstrate that ChannelViT not only surpasses the standard ViT in full-channel scenarios by enabling cross-channel reasoning but also exhibits superior robustness when channels are missing.
>   + **JUMP-CP:** The JUMP-CP benchmark, released by the Broad Institute, represents the largest public cell imaging dataset available at the time of our submission. While the mentioned Rxrx datasets and TissueNet are interesting, our comprehensive analysis on JUMP-CP covers similar ground. Also we were unable to access TissueNet via their link in the paper (https://netbio.bgu.ac.il/tissuenet3).
>   + **So2Sat:** Satellite imaging, with its multi-channel, multi-resolution data, is an ideal application for ChannelViT. Our empirical results confirm that ChannelViT outperforms ViT across various data splits and channel availability.
>   + **Camelyon17-WILDS:** Although Camelyon17 features fewer channels compared to JUMP-CP and So2Sat, the challenge of cross-hospital generalization remains compelling. ChannelViT's robust performance across varying staining color distributions from different hospitals is noteworthy. Due to space constraints, we have included this application in our appendix.
>   + Finally, we again appreciate the cell imaging references and will consider them for future work for life science journals. We hope this clarification addresses your concern regarding the scope of our paper.
> + **Model Scale:** We have expanded our results to include larger models: ViT-Base, ViT-Large, ChannelViT-Base, and ChannelViT-Large, detailed in Appendix C.7. Unsurprisingly, larger backbones result in superior performance. Our findings also reveal that despite the increased parameter count in ViT-Base and ViT-Large, ChannelViT maintains a significant performance lead. For instance, ViT-Large/16, with 13.9x more parameters, still lags behind ChannelViT-Small/16 (57.96 vs. 68.09), underscoring the importance of explicitly modeling cross channel interactions. We believe this addition should also address your concern regarding the efficiency of learning channel embeddings in larger ViT models.
> + **Time Efficiency:**
>   + We have added a new paragraph in Section 4.2 to discuss time efficiency, with a comprehensive runtime analysis presented in Table 8. Training ChannelViT is indeed 3.64x more resource-intensive, but the inference time is only 1.6x longer than ViT. We anticipate further efficiency gains by integrating ChannelViT with more efficient attention mechanisms like Linformer or LongNet.
>   + It is also important to highlight that the additional training time is offset by the avoidance of training separate expert models for each channel combination. ChannelViT's performance in handling inputs with varying channels results in considerable savings in practical applications.
>
> We trust that this rebuttal has thoroughly addressed the concerns you have raised. We welcome any further questions and thank you once again for your meticulous review.

---

> > ### Comment · Reviewer_ptju · 2023-11-22
> >
> > Dear authors,
> >
> > Thank you for your detailed response (and to other reviewers), and acknowledge reading them. In particular, I appreciate the experiments with CNNs (raised by Reviewer EcAm). Some important questions remain:
> >
> > 1. **Supervised ViT-S/16 Baselines**: I noticed that the results on ImageNet w/ ViT-S/16 DINO pretraining and linear probing (72.62% accuracy) are much lower than reported in the original DINO paper (77.0% accuracy). Performance for supervised ViT-S/16 (71.49% accuracy) is also much lower than what was reported in DINO (79.8% accuracy). With regularization, the performance of ViT-S/16 ranges from 76%-80% [1]. Likely, the linear probe evaluation of this work does not use recommended evaluation protocols for developing the supervised ViT-S/16 baseline, which undermines the authors claim on "ChannelViT not only surpasses the standard ViT in full-channel scenarios". Not considering suggested augmentation and regularization techniques may also harm comparisons with multi-channel evaluation. Would the authors be able to comment on this discrepancy?
> >
> > 2. **C17-WILDS over RxRx1-WILDS**: I acknowledge the author's comment on diversifying the evaluation of ChannelViT on different benchmarks (to not be specific to life sciences). However, the evaluation of C17-WILDS seems slightly out-of-place as despite being a microscopy imaging problem, it is ultimately not a multi-channel imaging problem unless one decides to separate the hematoxylin and eosin stains [2]. The channels do have distribution shift between the train and test set as a result of site-specific H&E stain intensity, but if choosing a microscopy benchmark from WILDS, why not evaluate RxRx1 which: 1) also has domain shift, and 2) is multi-channel? Overall, on choosing to evaluate tasks that showcase versatility and are out-of-domain of conventional natural image benchmarks, the reviewer thinks that the out-of-domain tasks could have been picked more appropriately to highlight ChannelViT's strengths.
> >
> > 3. **Existing C17-WILDS evaluation**: If choosing to evaluate on C17-WILDS, it is important to acknowledge existing baselines and leaderboards established that show +95% accuracy using ViT-B/16 [3] (though of little surprise as the baselines in [3] are with transfer learning). Coupled with the 1st point, the authors should contextualize their results with existing baselines, and clarify the experimental design around linear probing.
> >
> >
> > References
> > 1. Steiner, A., Kolesnikov, A., Zhai, X., Wightman, R., Uszkoreit, J. and Beyer, L., 2021. How to train your vit? data, augmentation, and regularization in vision transformers. arXiv preprint arXiv:2106.10270.
> > 2. https://scikit-image.org/docs/stable/auto_examples/color_exposure/plot_ihc_color_separation.html
> > 3. Kumar, A., Shen, R., Bubeck, S. and Gunasekar, S., 2022. How to fine-tune vision models with sgd. arXiv preprint arXiv:2211.09359.

---

### Official Review · Reviewer_PjTm · 2023-11-08

**Soundness:** 2 fair
**Presentation:** 3 good
**Contribution:** 2 fair
**Rating:** 5
**Confidence:** 4

**Summary:**

While the Vision Transformer has demonstrated robust performance with real-world images, its capacity to process multi-channel images, such as those from satellites, is somewhat constrained. To address this limitation, the authors have enhanced the conventional vision transformers by introducing a Hierarchical Channel Sampling (HCS) approach, which tackles the issue of sparse input channels. The proposed ChannelViT model has exhibited impressive results across three datasets, outperforming the traditional vision transformer.

**Strengths:**

+  The Hierarchical Channel Sampling (HCS) module enhances robustness by performing channel-wise sampling, which proves beneficial in scenarios involving incomplete image channels.
+  ChannelViT surpasses the conventional Vision Transformer (ViT) by demonstrating insensitivity to the number of input image channels, where ViT shows vulnerability.
+    A novel two-stage sampling algorithm is introduced within ChannelViT to selectively obscure input channels, optimizing the model's performance.

**Weaknesses:**

-    There is a potential risk of information loss, as highlighted in Section 3 of the methodology. The model's approach to segmenting the input image into various channel sequences and processing them individually could disrupt the alignment of channels, particularly in a 3-channel image prediction task.
-    The patch embedding technique described in Section 3.1 overlooks the issue of channel alignment when deconstructing images into separate channels.
-    The innovation of the proposed method warrants further scrutiny. Elements such as patch embedding, positional embedding, and Transformer Encoders, as discussed in Section 3.1, do not significantly diverge from the traditional Vision Transformer framework. These aspects should be acknowledged in the Related Work section, with a stronger emphasis on the unique contributions of this research.
-    The assertion in Section 3.2 that 'HCS guarantees equitable sampling across each m' lacks intuitive justification. While empirical results support this claim, a theoretical rationale would be beneficial.

**Questions:**

-    Is HCS also employed during the testing phase? It was mentioned that HCS simulates a test-time distribution during training.
-    Regarding Table 2, when selecting channels for testing, are these chosen at random, or were all possible combinations tested? If it's the latter, could you provide the mean and standard deviation of the results?
-    Has the model's performance been evaluated on datasets with varying channel availability, where some data might have only partial channels while other portions are fully channeled?

---

> ### Author Response · Authors · 2023-11-21
>
> Thank you for the detailed review and constructive suggestions. We have found them to be invaluable.
> + **Information Loss & Channel Alignment:** ChannelViT employs learnable positional and channel embeddings to align image patches effectively. The shared positional embeddings across channels enable ChannelViT to easily identify patch tokens from different channels at identical positions.
>   + **Empirical Evidence 1:** Figure 7 illustrates ChannelViT's ability to concentrate on the same region across RGB channels when identifying a zebra.
>   + **Empirical Evidence 2:** We conducted a sanity check to validate the alignment. By using ChannelViT's last layer output for each patch token to predict the original position and channel identity for each patch, we observed that ChannelViT achieves 100% accuracy on both ImageNet and JUMP-CP datasets within a single epoch of training. This underscores the effectiveness of the positional and channel embeddings.
> + **Distinguishing ChannelViT from Traditional ViT:** We appreciate your suggestion to delineate the differences between ChannelViT and the conventional ViT. We had initially discussed these distinctions in the second paragraph of our introduction. Following your advice, we have now included a comparative analysis in the related work and methodology sections of our revised manuscript.
>   + We would also like to emphasize that, although ChannelViT's patch and positional embeddings are conceptually similar to those in traditional ViT, several nuanced yet critical modifications have been made to enhance performance significantly.
>     + **Patch Embedding:** Traditional ViT acquires patch embeddings through a linear transformation of 3-channel image patches, with distinct weights for each channel. Conversely, ChannelViT processes each channel's image patches separately, employing shared weights across channels, which we demonstrate leads to improved performance (see Appendix C.3).
>     + **Positional Embedding:** Given that ChannelViT generates individual image patches for each channel, we opted to share positional embeddings, thereby equipping the model with cross-channel reasoning capabilities based on identical positions.
> + **Clarification of HCS in Section 3.2:**
>   + The variable m is uniformly sampled from the total number of channels (as described in step 1 of HCS), ensuring that the number of channels of the images processed by HCS are uniformly distributed over {1, 2, … C}. We have appended this clarification to the relevant paragraph to alleviate any confusion.
>   + We have clarified in Section 3.2 that HCS is utilized solely during the training phase.
> + **Evaluation in Table 2:** For an 8-channel image, we have considered all 255 possible channel combinations. The exhaustive results, including mean and standard deviation, are presented in Appendix C.8. Notably, despite each row having a consistent number of channels, the informational content across different channel combinations can vary markedly. This variation is reflected in the substantial standard deviation observed in Table 14. To further investigate this variance, we analyzed ChannelViT's performance gains over ViT for each channel combination, reporting both the mean improvement and standard deviation in Table 15. These findings affirm that ChannelViT's performance enhancements are not only uniform but also substantial.
> + **Evaluation on Datasets with Varying Channel Availability:**
>   + In Tables 3 and Figure 5, we have trained ViT and ChannelViT on datasets with diverse channel combinations, reporting their accuracy on both full and partial channels.
>   + In Tables 2 and 4, we trained ViT and ChannelViT on full channels and assessed their performance under various channel availability. We did not test them on datasets with a mix of partial and full channels. Since the models are fixed post-training, the evaluation scenario you described can be viewed as a blend of different channel combinations. Consequently, we can interpolate performance metrics from Table 2, which already encompasses all potential channel combinations.
>
> We trust that this rebuttal has addressed the concerns you have raised. Please let us know if you have other questions. Thank you once again for your thorough review.

---

> > ### Comment · Reviewer_PjTm · 2023-12-03
> > **Response to the comment**
> >
> > Thank you to the authors for their comprehensive responses to my previous queries. Here are some comments:
> >
> > - In 'Empirical Evidence 2', it's suggested that the model primarily requires training of the last layer for channel and positional information in one epoch. If this is the case, would it not be more efficient to validate the already trained model in your paper, rather than training a new model from scratch?
> > - In the section 'Distinguishing ChannelViT from Traditional ViT', the primary innovation appears to be the separation of embeddings for each channel. While the results are commendable, the scope of novelty in this approach seems somewhat limited.
> > - There seems to be a contradiction regarding the use of HCS, which is stated as being used only during training. Yet, elsewhere in the paper, HCS is mentioned in the context of test-time distribution during training. Clarification on this discrepancy would be beneficial.
> > - Both Reviewers EcAm and ptju have highlighted issues regarding training and inference time. Although the authors have provided detailed responses, concerns about time efficiency and resource intensity remain, especially considering the substantial computational resources required.
> > - Reviewer rJx2 has also pointed out the limited novelty of the work. Despite the authors’ assertion of strong performance, the overarching concern about the lack of novelty remains a critical aspect that needs addressing to enhance the paper's contribution to the field.
> >
> > Therefore, I would keep the initial rating.

---

### Author Response · Authors · 2023-11-21
**General reply**

We express our gratitude to the reviewers for their insightful comments and suggestions. In response, we have summarized the key contributions of our study and the key updates incorporated into the manuscript during the rebuttal period.

Contributions of this work:

+ **Robustness of ViTs to missing channels:** While previous studies have examined various robustness aspects of ViTs, including resilience to spurious correlations and adversarial perturbations, our research is the first to delve into the robustness of ViTs against missing channels. This is particularly relevant for multi-channel imaging applications where discrepancies in sensor data between the training and testing phases are prevalent.
+ **ChannelViT for multi-channel imaging:** Our proposed ChannelViT architecture generates patch tokens from individual channels, facilitating effective cross-channel and cross-positional reasoning. Distinct from existing approaches, ChannelViT introduces weight sharing among channels and incorporates learnable channel embeddings to capture channel-specific characteristics. Our empirical evaluations demonstrate that ChannelViT achieves significant performance improvements across four datasets and various baseline models (including ViT, MultiViT, FAN, and ResNets), while also enhancing the interpretability of the model.
+ **Hierarchical Channel Sampling:** Drawing inspiration from channel dropout, our proposed HCS regularizer not only improves training efficiency but also significantly enhances model robustness. HCS, even when applied to a standard ViT, results in markedly improved performance.

Main updates to the paper:
+ **Expanded baseline comparisons:** We have incorporated additional CNN baselines and two recently published ViT-based models in Appendix B for a more comprehensive comparison. Specifically, these additions include:
  + Convolutional Neural Networks (CNNs) with ResNet-50 and ResNet-152 architectures.
  + MultiViTs that utilize separate ViTs for each input channel, and integrate their outputs for the final prediction [1].
  + Fully Attentional Networks (FANs) that integrate channel-wise self-attention within the transformer block [2].
+ **Evaluation of models at larger scales:** The results for ViT-Base, ViT-Large, ChannelViT-Base, and ChannelViT-Large have been included in Appendix C.7 to demonstrate the performance scalability.
+ **Time efficiency analysis:* A new section (Time efficiency in Section 4.2) has been added to the main body of the paper to address the time efficiency of ChannelViT. A more in-depth analysis of training and inference times are provided in Appendix C.1.

While the aforementioned points highlight the main updates, we have also made smaller adjustments throughout the paper to meticulously address the specific issues raised by the reviewers. We believe these updates comprehensively address the reviewers' feedback and substantiate the contributions of our work to the field.

References
1. Hussein, Ramy, Soojin Lee, and Rabab Ward. "Multi-channel vision transformer for epileptic seizure prediction." Biomedicines 10.7 (2022): 1551.
2. Zhou, Daquan, et al. "Understanding the robustness in vision transformers." International Conference on Machine Learning. PMLR, 2022.

---

### Meta-Review · Area_Chair_3Pae · 2023-12-11

**Metareview:**

The paper presents ChannelViT, an extension of the Vision Transformer (ViT) architecture tailored for multi-channel image analysis. It incorporates a novel tokenization scheme, splitting images into patches across spatial and channel dimensions.

The reviewers gave mixed feedback regarding the paper, with ratings of 8, 8, 5, 5.
  Some reviewers found the proposed approach to multi-channel image analysis to be simple yet effective, with the potential for unique applications in multiplexed imaging. Reviewers also appreciated the extensive experimentation and analysis across multiple datasets.
  However, several key concerns were raised regarding the novelty of the work, with some reviewers suggesting that the channel splitting and HCS could be viewed as marginal extensions of existing ideas. Some reviewers also highlighted potential issues with computational cost, efficiency, and the relevance of the chosen datasets for the paper's targeted application in multiplex imaging. There was also a call for additional baselines and comparisons with state-of-the-art methods in the relevant domain.

The authors responded to these concerns by emphasizing the novel aspects of their work, such as the focus on channel robustness and the introduction of weight sharing across channels. They also provided some additional experimental results and comparisons with recent ViT variants for multi-channel imaging.

After reviewing all the feedback and comments,  the AC recommends acceptance of the paper, subject to the incorporation of all the new experimental results and the future addition of experiments and comparisons.

**Justification For Why Not Higher Score:**

The paper holds potential interest for the broader community; however, several significant issues listed above have prevented the AC from assigning a higher score.

**Justification For Why Not Lower Score:**

Though the paper received mixed scores (two reviewers gave 5), the ACs still think it's valuable to share and the result is promising.

---

### Decision · Program_Chairs · 2024-01-16

Accept (poster)